# DIMES: A Differentiable Meta Solver for Combinatorial Optimization Problems

**Ruizhong Qiu**[*][†]
Department of Computer Science
University of Illinois Urbana–Champaign
rq5@illinois.edu

**Zhiqing Sun**[*]**, Yiming Yang**
Language Technologies Institute
Carnegie Mellon University
{zhiqings,yiming}@cs.cmu.edu

## Abstract

Recently, deep reinforcement learning (DRL) models have shown promising results in solving NP-hard Combinatorial Optimization (CO) problems. However, most DRL solvers can only scale to a few hundreds of nodes for combinatorial optimization problems on graphs, such as the Traveling Salesman Problem (TSP). This paper addresses the scalability challenge in large-scale combinatorial optimization by proposing a novel approach, namely, DIMES. Unlike previous DRL methods which suffer from costly autoregressive decoding or iterative refinements of discrete solutions, DIMES introduces a compact continuous space for parameterizing the underlying distribution of candidate solutions. Such a continuous space allows stable REINFORCE-based training and fine-tuning via massively parallel sampling. We further propose a meta-learning framework to enable effective initialization of model parameters in the fine-tuning stage. Extensive experiments show that DIMES outperforms recent DRL-based methods on large benchmark datasets for Traveling Salesman Problems and Maximal Independent Set problems.

## 1 Introduction

Combinatorial Optimization (CO) is a fundamental problem in computer science. It has important real-world applications such as shipment planning, transportation, robots routing, biology, circuit design, and more [67]. However, due to NP-hardness, a significant portion of the CO problems suffer from an exponential computational cost when using traditional algorithms. As a well-known example, the Traveling Salesman Problem (TSP) has been intensively studied [35, 60] for finding the most cost-effective tour over an input graph where each node is visited exactly once before finally returning to the start node. Over the past decades, significant effort has been made for designing more efficient heuristic solvers [5, 20] to approximate near-optimal solutions in a reduced search space.

Recent development in deep reinforcement learning (DRL) has shown promises in solving CO problems without manual injection of domain-specific expert knowledge [7, 42, 45]. The appeal of neural methods is because they can learn useful patterns (such as graph motifs) from data, which might be difficult to discover by hand. A typical category of DRL solvers, namely *construction heuristics learners*, [7, 42] uses a Markov decision process (MDP) to grow partial solutions by adding one new node per step, with a trained strategy which assigns higher probabilities to better solutions. Another category of DRL-based solvers, namely *improvement heuristics learners* [10, 72], iteratively refines a feasible solution with neural network-guided local Operations Research (OR) operations [64]. A major limitation of these DRL solvers lies in their scalability on large instances. For example, current DRL solvers for TSP can only scale to graphs with up to hundreds of nodes.

---

[*]Equal contribution.
[†]Work was done during internship at CMU.

36th Conference on Neural Information Processing Systems (NeurIPS 2022).

The bad scalability of these DRL methods lies in the fact that they suffer from costly decoding of CO solutions, which is typically linear in the number of nodes in the input graph. Since the reward of reinforcement learning is determined after decoding a complete solution (with either a chain rule factorization or iterative refinements), either construction or improvement heuristic learners would encounter the sparse reward problem when dealing with large graphs [42, 33, 37]. While such an overhead can be partially alleviated by constructing several parts of the solution in parallel [1] for locally decomposable CO problems[3], such as for maximum independent set (MIS) problems [54], how to scale up neural solvers for CO problems in general, including the locally non-decomposable ones (such as TSP) is still an open challenge.

In this paper, we address the scalability challenge by proposing a novel framework, namely DIMES (DIfferentiable MEta Solver), for solving combinatorial optimization problems. Unlike previous DRL-based CO solvers that rely on construction or improvement heuristics, we introduce a compact continuous space to parameterize the underlying distribution of candidate solutions, which allows massively parallel on-policy sampling without the costly decoding process, and effectively reduces the variance of the gradients by the REINFORCE algorithm [71] during both training and fine-tuning phases. We further propose a meta-learning framework for CO over problem instances to enable effective initialization of model parameters in the fine-tuning stage. To our knowledge, we are the first to apply meta-learning over a collection of CO problem instances, where each instance graph is treated as one of a collection tasks in a unified framework.

We need to point out that the idea of designing a continuous space for combinatorial optimization problems has been tried by the heatmaps approaches in the literature [48, 31, 19, 14, 44]. However, there are major distinctions between the existing methods and our DIMES. For instance, Fu et al. [19] learn to generate heatmaps via supervised learning (i.e., each training instance is paired with its best solution) [4, 21], which is very costly to obtain on large graphs. DIMES is directly optimized with gradients estimated by the REINFORCE algorithm without any supervision, so it can be trained on large graphs directly. As a result, DIMES can scale to large graphs with up to tens of thousands of nodes, and predict (nearly) optimal solutions without the need for costly generation of supervised training data or human specification of problem-specific heuristics.

In our experiments, we show that DIMES outperforms strong baselines among DRL-based solvers on TSP benchmark datasets, and can successfully scale up to graphs with tens of thousands of nodes. As a sanity check, we also evaluate our framework with locally decomposable combinatorial optimization problems, including Maximal Independent Set (MIS) problem for synthetic graphs and graphs reduced from satisfiability (SAT) problems. Our experimental results show that DIMES achieve competitive performance compared to neural solvers specially designed for locally decomposable CO problems.

## 2 Related Work

### 2.1 DRL-Based Construction Heuristics Learners

Construction heuristics methods create a solution of CO problem instance in one shot without further modifications. Bello et al. [7] are the first to tackle combinatorial optimization problems using neural networks and reinforcement learning. They used a Pointer Network (PtrNet) [68] as the policy network and used the actor-critic algorithm [41] for training on TSP and KnapSack instances. Further improved models have been developed afterwards [13, 42, 63, 14, 46], such as attention models [66], better DRL algorithms [36, 52, 43, 45, 59, 73, 69], for an extended scope of CO problems such as Capacitated Vehicle Routing Problem (CVRP) [57], Job Shop Scheduling Problem (JSSP) [76], Maximal Independent Set (MIS) problem [36, 1], and boolean satisfiability problem (SAT) [75].

Our proposed method in this paper belongs to the category of *construction heuristics learners* in the sense of producing a one-shot solution per problem instance. However, there are major distinctions between previous methods and ours. One distinction is how to construct solutions. Unlike previous methods which generate the solutions via a constructive Markov decision process (MDP) with rather costly decoding steps (adding one un-visited node per step to a partial solution), we introduce a compact continuous space to parameterize the underlying distribution of discrete candidate solutions, and to allow efficient sampling from that distribution without costly neural network-involved decoding.

---

[3]Locally decomposable problem refers to the problem where the feasibility constraint and the objective can be decomposed by locally connected variables (in a graph) [1].

Another distinction is about the training framework. For instance, Drori et al. [14] proposes a similar solution decoding scheme but employs a DRL framework to train the model. Instead, we propose a much more effective meta-learning framework to train our model, enabling DIMES to be trained on large graphs directly.

## 2.2 DRL-Based Improvement Heuristics Learners

In contrast to construction heuristics, DRL-based improvement heuristics methods train a neural network to iteratively improve the quality of the current solution until computational budget runs out. Such DRL-based improvement heuristics methods are usually inspired by classical local search algorithms such as 2-opt [11] and the large neighborhood search (LNS) [65], and have been demonstrated with outstanding results by many previous work [72, 51, 70, 12, 10, 26, 74, 53, 29, 37]. Improvement heuristics methods generally show better performance than construction heuristics methods but are slower in computation in return.

## 2.3 Supervised Learners for CO Problems

Vinyals et al. [68] trained a Pointer Network to predict a TSP solution based on supervision signals from the Held–Karp algorithm [6] or approximate algorithms. Li et al. [48] and Joshi et al. [31] trained a graph convolutional network to predict the possibility of each node or edge to be included in the optimal solutions of MIS and TSP problems, respectively. Recently, Joshi et al. [32] showed that unsupervised reinforcement learning leads to better emergent generalization over various sized graphs than supervised learning. Our work in this paper provides further evidence for the benefits of the unsupervised training, or more specifically, unsupervised generation of heatmaps [48, 31, 19, 44], for combinatorial optimization problems.

# 3 Proposed Method

## 3.1 Formal Definitions

Following a conventional notation [61] we define $\mathcal{F}_s$ as the set of discrete feasible solutions for a CO problem instance $s$, and $c_s : \mathcal{F}_s \to \mathbb{R}$ as the cost function for feasible solutions $f \in \mathcal{F}_s$. The objective is to find the optimal solution for a given instance $s$:

$$f_s^* = \operatorname*{argmin}_{f \in \mathcal{F}_s} c_s(f). \tag{1}$$

For the Traveling Salesman Problem (TSP), $\mathcal{F}_s$ is the set of all the tours that visit each node exactly once and returns to the starting node at the end, and $c_s$ calculates the cost for each tour $f \in \mathcal{F}_s$ by summing up the edge weights in the tour. The size of $\mathcal{F}_s$ for TSP is $n!$ for a graph with $n$ nodes. For the Maximal Independent Set (MIS) problem, $\mathcal{F}_s$ is a subset of the power set $\mathcal{S}_s = \{0, 1\}^n$ and consists of all the independent subsets where each node of a subset has no connection to any other node in the same subset, and $c_s$ calculates the negation of the size of each independent subset.

We parameterize the solution space with a continuous and differentiable vector $\boldsymbol{\theta} \in \mathbb{R}^{|\mathcal{V}_s|}$, where $\mathcal{V}_s$ denotes the variables in the problem instance $s$ (e.g., edges in TSP and nodes in MIS), and estimates the probability of each feasible solution $f$ as:

$$p_{\boldsymbol{\theta}}(f \mid s) \propto \exp\left(\sum_{i=1}^{|\mathcal{V}_s|} f_i \cdot \theta_i\right) \quad \text{subject to} \quad f \in \mathcal{F}_s. \tag{2}$$

where $p_{\boldsymbol{\theta}}$ is an energy function over the discrete feasible solution space, $f$ is a $|\mathcal{V}_s|$-dimensional vector with element $f_i \in \{0, 1\}$ indicating whether the $i^{\text{th}}$ variable is included in feasible solution $f$, and the higher value of $\theta_i$ means a higher probability for the $i^{\text{th}}$ variable produced by $p_{\boldsymbol{\theta}}(f \mid s)$.

## 3.2 Gradient-Based Optimization

When the combinatorial problem is locally decomposable, such a MIS, a penalty loss [34, 2] can be added to suppress the unfeasible solutions, e.g.:

$$\ell_{\text{Erdős}}(\boldsymbol{\theta} \mid s) = \sum_{f \in \mathcal{S}_s} \left[ p_{\boldsymbol{\theta}}(f \mid s) \cdot (c_s(f) + \beta \cdot \mathbb{1}(f \notin \mathcal{F}_s)) \right]. \tag{3}$$

where $\beta > \max_{f \in \mathcal{F}_s} c_s(f)$. The objective function $\ell_{\text{Erdős}}$ can thus be calculated analytically and enable end-to-end training. However, this is not always possible for general structured combinatorial problems such as TSP[4]. Therefore, we propose to directly optimize the expected cost over the underlying population of feasible solutions, which is defined as:

$$\ell_p(\boldsymbol{\theta} \mid s) = \mathbb{E}_{f \sim p_{\boldsymbol{\theta}}}[c_s(f)]. \tag{4}$$

Optimizing this objective requires efficient sampling, with which REINFORCE-based [71] gradient estimation can be calculate. Nevertheless, a common practice to sample from the energy $p_{\boldsymbol{\theta}}$ functions requires MCMC [47], which is not efficient enough. Hence we propose to design an auxiliary distribution $q_{\boldsymbol{\theta}}$ over the feasible solutions $\mathcal{F}_s$, such that the following conditions hold: 1) sampling from $q_{\boldsymbol{\theta}}$ is efficient, and 2) $q_{\boldsymbol{\theta}}$ and $p_{\boldsymbol{\theta}}$ should convergence to the same optimal $\boldsymbol{\theta}^*$. Then, we can replace $p_{\boldsymbol{\theta}}$ by $q_{\boldsymbol{\theta}}$ in our objective function as:

$$\ell_q(\boldsymbol{\theta} \mid s) = \mathbb{E}_{f \sim q_{\boldsymbol{\theta}}}[c_s(f)], \tag{5}$$

and get the REINFORCE-based update rule as:

$$\nabla_{\boldsymbol{\theta}} \mathbb{E}_{f \sim q_{\boldsymbol{\theta}}}[c_s(f)] = \mathbb{E}_{f \sim q_{\boldsymbol{\theta}}}[(c_s(f) - b(s))\nabla_{\boldsymbol{\theta}} \log q_{\boldsymbol{\theta}}(f)], \tag{6}$$

where $b(s)$ denotes a baseline function that does not depend on $f$ and estimates the expected cost to reduce the variance of the gradients. In this paper, we use a sampling-based baseline function proposed by Kool et al. [43].

Next, we specify the auxiliary distributions for TSP and MIS, respectively. For brevity, we omit the conditional notations of $s$ for all probability formulas in the rest of the paper.

### 3.2.1 Auxiliary Distribution for TSP

For TSP on an $n$-node graph, each feasible solution $f$ consists of $n$ edges forming a tour, which can be specified as a permutation $\pi_f$ of $n$ nodes, where $\pi_f(0) = \pi_f(n)$ is the start/end node, and $\pi_f(i) \neq \pi_f(j)$ for any $i, j$ with $0 \leq i, j < n$ and $i \neq j$. Note that for a single solution $f$, $n$ different choices of the start node $\pi_f(0)$ correspond to $n$ different permutations $\pi_f$. In this paper, we choose the start node $\pi_f(0)$ randomly with a uniform distribution:

$$q_{\text{TSP}}(\pi_f(0) = j) := \frac{1}{n} \quad \text{for any node } j; \tag{7}$$

$$q_{\boldsymbol{\theta}}^{\text{TSP}}(f) := \sum_{j=0}^{n-1} \frac{1}{n} \cdot q_{\text{TSP}}(\pi_f \mid \pi_f(0) = j). \tag{8}$$

Given the start node $\pi_f(0)$, we factorize the probability via chain rule in the visiting order:

$$q_{\text{TSP}}(\pi_f \mid \pi_f(0)) := \prod_{i=1}^{n-1} q_{\text{TSP}}(\pi_f(i) \mid \pi_f(< i)). \tag{9}$$

Since the variables in TSP are edges, we let $\theta_{i,j}$ denote the $\theta$ value of edge from node $i$ to node $j$ for notational simplicity, i.e., we use a matrix $\boldsymbol{\theta} \in \mathbb{R}^{n \times n}$ to parameterize the probabilistic distribution of $n!$ discrete feasible solutions. We define:

$$q_{\text{TSP}}(\pi_f(i) \mid \pi_f(< i)) := \frac{\exp(\theta_{\pi_f(i-1), \pi_f(i)})}{\sum_{j=i}^{n} \exp(\theta_{\pi_f(i-1), \pi_f(j)})}. \tag{10}$$

Here a higher valued $\theta_{i,j}$ corresponds to a higher probability for the edge from node $i$ to node $j$ to be sampled. The compact, continuous and differentiable space of $\boldsymbol{\theta}$ allows us to leverage gradient-based optimization without costly MDP-based construction of feasible solutions, which has been a bottleneck for scaling up in representative DRL solvers so far. In other words, we also no longer need costly MCMC-based sampling for optimizing our model due to the chain-rule decomposition. Instead, we use autoregressive factorization for sampling from the auxiliary distribution, which is faster than sampling with MCMC from the distribution defined by the energy function.

---

[4]TSP has a global constraint of forming a Hamiltonian cycle.

### 3.2.2 Auxiliary Distribution for MIS

For the Maximal Independent Set (MIS) problem, the feasible solution is a set of independent nodes, which means that none of the node has any link to any other node in the same set. To ease the analysis, we further impose a constraint to the MIS solutions such that each set is not a proper subset of any other independent set in the feasible domain.

To enable the chain-rule decomposition in probability estimation, we introduce $\boldsymbol{a}$ as an ordering of the independent nodes in solution $f$, and $\{\boldsymbol{a}\}_f$ as the set of all possible orderings of the nodes in $f$. The chain rule applied to $\boldsymbol{a}$ can thus be defined as:

$$q_{\boldsymbol{\theta}}^{\mathrm{MIS}}(f) = \sum_{\boldsymbol{a} \in \{\boldsymbol{a}\}_f} q_{\mathrm{MIS}}(\boldsymbol{a}), \tag{11}$$

$$q_{\mathrm{MIS}}(\boldsymbol{a}) = \prod_{i=1}^{|\boldsymbol{a}|} q_{\mathrm{MIS}}(a_i \mid \boldsymbol{a}_{<i}) = \prod_{i=1}^{|\boldsymbol{a}|} \frac{\exp(\theta_{a_i})}{\sum_{j \in \mathcal{G}(\boldsymbol{a}_{<i})} \exp(\theta_j)}.$$

where $\mathcal{G}(\boldsymbol{a}_{<i})$ denotes the set of available nodes for growing partial solution $(a_1, \ldots, a_{i-1})$, i.e., the nodes that have no edge to any nodes in $\{a_1, \ldots, a_{i-1}\}$. Notice again that the parameterization space for MIS $\boldsymbol{\theta} \in \mathbb{R}^n$ (where $n$ denotes the number of nodes in the graph) is compact, continuous and differentiable, which allows efficient gradient-driven optimization.

Due to the space limit, we leave the proof of the convergence between $p_{\boldsymbol{\theta}}$ and $q_{\boldsymbol{\theta}}$ (i.e., $q_{\boldsymbol{\theta}}^{\mathrm{TSP}}$ and $q_{\boldsymbol{\theta}}^{\mathrm{MIS}}$) to the appendix.

### 3.3 Meta-Learning Framework

Model-Agnostic Meta-Learning (MAML) [18] is originally proposed for few-shot learning. In the MAML framework, a model is first trained on a collection of *tasks* simultaneously, and then adapts its model parameters to each task. The standard MAML uses second-order derivatives in training, which are costly to compute. To reduce computation burden, the authors also propose first-order approximation that does not require second-order derivatives.

Inspired by MAML, we train a graph neural network (GNN) over a collection of *problem instances* in a way that the it can capture the common nature across all the instances, and adapt its distribution parameters effectively to each instance based on the features/structure of each input graph. Let $F_{\boldsymbol{\Phi}}$ be the graph neural network with parameter $\boldsymbol{\Phi}$, and denote by $\boldsymbol{\kappa}_s$ the input features of an instance graph $s$ in collection $\mathcal{C}$, by $\boldsymbol{A}_s$ the adjacency matrix of the input graph, and by $\boldsymbol{\theta}_s := F_{\boldsymbol{\Phi}}(\boldsymbol{\kappa}_s, \boldsymbol{A}_s)$ the instance-specific initialization of distribution parameters. The vanilla loss function is defined as the expected cost of the solution for any graph in the collection as:

$$\mathcal{L}(\boldsymbol{\Phi} \mid \mathcal{C}) = \mathbb{E}_{s \in \mathcal{C}} \ell_q(\boldsymbol{\theta}_s) = \mathbb{E}_{s \in \mathcal{C}} \ell_q(F_{\boldsymbol{\Phi}}(\boldsymbol{\kappa}_s, \boldsymbol{A}_s)). \tag{12}$$

The gradient-based updates can thus be written as:

$$\begin{aligned} \nabla_{\boldsymbol{\Phi}} \mathcal{L}(\boldsymbol{\Phi} \mid \mathcal{C}) &= \mathbb{E}_{s \in \mathcal{C}} \left[ \nabla_{\boldsymbol{\Phi}} \boldsymbol{\theta}_s \cdot \nabla_{\boldsymbol{\theta}_s} \ell_q(\boldsymbol{\theta}_s) \right] \\ &= \mathbb{E}_{s \in \mathcal{C}} \left[ \nabla_{\boldsymbol{\Phi}} F_{\boldsymbol{\Phi}}(\boldsymbol{\kappa}_s, \boldsymbol{A}_s) \cdot \nabla_{\boldsymbol{\theta}_s} \ell_q(\boldsymbol{\theta}_s) \right]. \end{aligned} \tag{13}$$

where $\nabla_{\boldsymbol{\theta}_s} \ell_q(\boldsymbol{\theta}_s)$ is estimated using the REINFORCE algorithm (Equation 6). Since $\ell_q$ does not depend on the ground-truth labels, we can further fine-tune neural network parameters on each single test instance with REINFORCE-based updates, which is referred to as *active search* [7, 28].

Specifically, the fine-tuned parameters $\boldsymbol{\Phi}_s^{(T)}$ is computed using one or more gradient updates for each graph instance $s$. For example, when adapting to a problem instance $s$ using $T$ gradient updates with learning rate $\alpha$, we have:

$$\boldsymbol{\Phi}_s^{(0)} = \boldsymbol{\Phi}, \qquad \boldsymbol{\Phi}_s^{(t)} = \boldsymbol{\Phi}_s^{(t-1)} - \alpha \nabla_{\boldsymbol{\Phi}_s^{(t-1)}} \mathcal{L}(\boldsymbol{\Phi}_s^{(t-1)} \mid \{s\}) \quad \text{for} \quad 1 \le t \le T, \tag{14}$$

$$\boldsymbol{\theta}_s^{(T)} = F_{\boldsymbol{\Phi}_s^{(T)}}(\boldsymbol{\kappa}_s, \boldsymbol{A}_s). \tag{15}$$

Here we use AdamW [50] in our experiments. Next, we optimize the performance of the graph neural network with updated parameters (i.e., $\boldsymbol{\Phi}_s^{(T)}$) with respect to $\boldsymbol{\Phi}$, with a meta-objective:

$$\mathcal{L}_{\mathrm{meta}}(\boldsymbol{\Phi} \mid \mathcal{C}) = \mathbb{E}_{s \in \mathcal{C}} \ell_q(\boldsymbol{\theta}_s^{(T)} \mid s), \tag{16}$$

---

**Algorithm 1** MAML in DIMES

---

**Input:** $p(\mathcal{C})$: distribution over CO problem instances
**Input:** $\alpha$: step size hyperparameters
 1: randomly initialize $\boldsymbol{\Phi}$
 2: **while** not done **do**
 3:     Sample batch of graph instances $c_i \sim p(\mathcal{C})$
 4:     **for all** $c_i$ **do**
 5:         Sample $K$ solutions $\mathcal{D}_i = \{f_1, f_2, \ldots, f_K\}$ using $q_{F_{\boldsymbol{\Phi}}(\boldsymbol{\kappa}_s, \boldsymbol{A}_s)}$ for $c_i$
 6:         Evaluate $\nabla_{\boldsymbol{\Phi}} \ell_q(F_{\boldsymbol{\Phi}}(\boldsymbol{\kappa}_s, \boldsymbol{A}_s))$ using $\mathcal{D}$ in Equation 13
 7:         Compute adapted parameters with Equation 14: $\boldsymbol{\Phi}_i^{(T)} = \text{GradDescent}^{(T)}(\boldsymbol{\Phi})$
 8:         Sample $K$ solutions $\mathcal{D}_i' = \{f_1', f_2', \ldots, f_K'\}$ using $q_{F_{\boldsymbol{\Phi}_s^{(T)}}(\boldsymbol{\kappa}_s, \boldsymbol{A}_s)}$ for $c_i$
 9:     **end for**
10:     Update $\boldsymbol{\Phi} = \boldsymbol{\Phi} - \text{AdamW}\big(\sum_{c_i \in p(\mathcal{C})} \nabla_{\boldsymbol{\Phi}} \ell_q(F_{\boldsymbol{\Phi}_s^{(T)}}(\boldsymbol{\kappa}_s, \boldsymbol{A}_s))\big)$ using each $\mathcal{D}_i'$ in Equation (17)
11: **end while**

---

and calculate the meta-updates as:

$$
\begin{aligned}
\nabla_{\boldsymbol{\Phi}} \mathcal{L}_{\text{meta}}(\boldsymbol{\Phi} \mid \mathcal{C}) &= \mathbb{E}_{s \in \mathcal{C}} \left[ \nabla_{\boldsymbol{\Phi}} \boldsymbol{\theta}_s^{(T)} \cdot \nabla_{\boldsymbol{\theta}_s^{(T)}} \ell_q(\boldsymbol{\theta}_s^{(T)}) \right] \\
&\approx \mathbb{E}_{s \in \mathcal{C}} \left[ \nabla_{\boldsymbol{\Phi}_s^{(T)}} F_{\boldsymbol{\Phi}_s^{(T)}}(\boldsymbol{\kappa}_s, \boldsymbol{A}_s) \cdot \nabla_{\boldsymbol{\theta}_s^{(T)}} \ell_q(\boldsymbol{\theta}_s^{(T)}) \right].
\end{aligned}
\tag{17}
$$

Notice that we adopt the first-order approximation to optimize this objective, which ignores the update via the gradient term of $\nabla_{\boldsymbol{\Phi}} \mathcal{L}(\boldsymbol{\Phi} \mid \{s\})$. We defer the derivation of the approximation formula to the appendix. Algorithm 1 illustrates the full training process of our meta-learning framework.

### 3.4 Per-Instance Search

Given a fine-tuned (i.e., after active search) continuous parameterization of the solution space $\boldsymbol{\theta}_s^{(T)}$, the per-instance search decoding aims to search for a feasible solution that minimizes the cost function $c$. In this paper, we adotp three decoding strategies, i.e., greedy decoding, sampling, and Monte Carlo tree search. Due to the space limit, the detailed description of three decoding strategies can be found in the appendix.

### 3.5 Graph Neural Networks

Based on the shape of the differentiable variable $\boldsymbol{\theta}$ required by each problem (i.e., $\mathbb{R}^{n \times n}$ for TSP and $\mathbb{R}^n$ for MIS), we use Anisotropic Graph Neural Networks [9] and Graph Convolutional Networks [40] as the backbone network for TSP and MIS tasks, respectively. Due to the space limit, the detailed neural architecture design can be found in the appendix.

## 4 Experiments

### 4.1 Experiments for Traveling Salesman Problem

#### 4.1.1 Experimental Settings

**Data Sets**   The training instances are generated on the fly. We closely follow the data generation procedure of previous works, e.g., [42]. We generate 2-D Euclidean TSP instances by sampling each node independently from a uniform distribution over the unit square. The TSP problems of different scales are named TSP-500/1000/10000, respectively, where TSP-$n$ indicates the TSP instance on $n$ nodes. For testing, we use the test instances generated by Fu et al. [19]. There are 128 test instances in each of TSP-500/1000, and 16 test instances in TSP-10000.

**Evaluation Metrics**   For model comparison, we report the average length (Length), average performance drop (Drop) and averaged inference latency time (Time), respectively, where *Length* (the shorter, the better) is the average length of the system-predicted tour for each test-set graph, *Drop* (the smaller, the better) is the average of relative performance drop in terms of the solution length compared to a baseline method, and *Time* (the smaller, the better) is the total clock time for generating solutions for all test instance, in seconds (s), minutes (m), or hours (h).

Table 1: Results of TSP. See Section 4.1.2 for detailed descriptions. * indicates the baseline for computing the performance drop. Results of baselines (except those of EAS and the running time of LKH-3, POMO, and Att-GCN) are taken from Fu et al. [19].

| Method | Type | TSP-500 | | | TSP-1000 | | | TSP-10000 | | |
|---|---|---|---|---|---|---|---|---|---|---|
| | | Length ↓ | Drop ↓ | Time ↓ | Length ↓ | Drop ↓ | Time ↓ | Length ↓ | Drop ↓ | Time ↓ |
| Concorde | OR (exact) | 16.55* | — | 37.66m | 23.12* | — | 6.65h | N/A | N/A | N/A |
| Gurobi | OR (exact) | 16.55 | 0.00% | 45.63h | N/A | N/A | N/A | N/A | N/A | N/A |
| LKH-3 (default) | OR | 16.55 | 0.00% | 46.28m | 23.12 | 0.00% | 2.57h | 71.77* | — | 8.8h |
| LKH-3 (less trails) | OR | 16.55 | 0.00% | 3.03m | 23.12 | 0.00% | 7.73m | 71.79 | — | 51.27m |
| Nearest Insertion | OR | 20.62 | 24.59% | 0s | 28.96 | 25.26% | 0s | 90.51 | 26.11% | 6s |
| Random Insertion | OR | 18.57 | 12.21% | 0s | 26.12 | 12.98% | 0s | 81.85 | 14.04% | 4s |
| Farthest Insertion | OR | 18.30 | 10.57% | 0s | 25.72 | 11.25% | 0s | 80.59 | 12.29% | 6s |
| EAN | RL+S | 28.63 | 73.03% | 20.18m | 50.30 | 117.59% | 37.07m | N/A | N/A | N/A |
| EAN | RL+S+2-OPT | 23.75 | 43.57% | 57.76m | 47.73 | 106.46% | 5.39h | N/A | N/A | N/A |
| AM | RL+S | 22.64 | 36.84% | 15.64m | 42.80 | 85.15% | 63.97m | 431.58 | 501.27% | 12.63m |
| AM | RL+G | 20.02 | 20.99% | 1.51m | 31.15 | 34.75% | 3.18m | 141.68 | 97.39% | 5.99m |
| AM | RL+BS | 19.53 | 18.03% | 21.99m | 29.90 | 29.23% | 1.64h | 129.40 | 80.28% | 1.81h |
| GCN | SL+G | 29.72 | 79.61% | 6.67m | 48.62 | 110.29% | 28.52m | N/A | N/A | N/A |
| GCN | SL+BS | 30.37 | 83.55% | 38.02m | 51.26 | 121.73% | 51.67m | N/A | N/A | N/A |
| POMO+EAS-Emb | RL+AS | 19.24 | 16.25% | 12.80h | N/A | N/A | N/A | N/A | N/A | N/A |
| POMO+EAS-Lay | RL+AS | 19.35 | 16.92% | 16.19h | N/A | N/A | N/A | N/A | N/A | N/A |
| POMO+EAS-Tab | RL+AS | 24.54 | 48.22% | 11.61h | 49.56 | 114.36% | 63.45h | N/A | N/A | N/A |
| Att-GCN | SL+MCTS | 16.97 | 2.54% | 2.20m | 23.86 | 3.22% | 4.10m | 74.93 | 4.39% | 21.49m |
| | RL+G | 18.93 | 14.38% | 0.97m | 26.58 | 14.97% | 2.08m | 86.44 | 20.44% | 4.65m |
| | RL+AS+G | 17.81 | 7.61% | 2.10h | 24.91 | 7.74% | 4.49h | 80.45 | 12.09% | 3.07h |
| DIMES (ours) | RL+S | 18.84 | 13.84% | 1.06m | 26.36 | 14.01% | 2.38m | 85.75 | 19.48% | 4.80m |
| | RL+AS+S | 17.80 | 7.55% | 2.11h | 24.89 | 7.70% | 4.53h | 80.42 | 12.05% | 3.12h |
| | RL+MCTS | 16.87 | 1.93% | 2.92m | 23.73 | 2.64% | 6.87m | 74.63 | 3.98% | 29.83m |
| | RL+AS+MCTS | **16.84** | **1.76%** | 2.15h | **23.69** | **2.46%** | 4.62h | **74.06** | **3.19%** | 3.57h |

Table 2: Ablation study on TSP-1000.

(a) On meta-learning ($T = 10$).

| Inner updates | Fine-tuning | Length ↓ |
|---|---|---|
| | | 27.11 |
| ✓ | | 26.58 |
| | ✓ | 25.68 |
| ✓ | ✓ | **24.91** |

(b) On fine-tuning parts ($T = 5$).

| Part | Length ↓ |
|---|---|
| Cont. Param. | 27.73 |
| MLP | 26.75 |
| GNNOut+MLP | **26.49** |
| GNN+MLP | 26.81 |

(c) On inner update steps $T$.

| $T$ | 0 | 4 | 8 | 10 | 12 | 14 |
|---|---|---|---|---|---|---|
| Length ↓ | 25.79 | 25.28 | 25.08 | 25.08 | 24.97 | 24.91 |

(d) On heatmaps for MCTS.

| Heatmap | Length ↓ |
|---|---|
| $\mathrm{Unif}(0, 1)$ | 25.52 |
| $1/(r_i + 1)$ | 24.14 |
| Att-GCN | 23.86 |
| DIMES (ours) | **23.69** |

**Training and Hardware**  Due to the space limit, please refer to the appendix.

### 4.1.2 Main Results

Our main results are summarized in Table 1, with $T = 15$ for TSP-500, $T = 14$ for TSP-1000, and $T = 12$ for TSP-10000. We use a GNN followed by an MLP as the backbone, whose detailed architecture is defered to the appendix. Note that we fine-tune the GNN output and the MLP only. For the evaluation of DIMES, we fine-tune the DIMES on each instance for 100 steps (TSP-500 & TSP-1000) or for 50 steps (TSP-10000). For the sampling in DIMES, we use the temperature parameter $\tau = 0.01$ for DIMES+S and $\tau = 1$ for DIMES+AS+S. We compare DIMES with 14 other TSP solvers on the same test sets. We divide those 14 methods into two categories: 6 traditional OR methods and 8 learning-based methods.

- Traditional operations research methods include two exact solvers, i.e., Concorde [4] and Gurobi [21], and a strong heuristic solver named LKH-3 [23]. For LKH-3, we consider two settings: (i) *default*: following previous work [42], we perform 1 runs with a maximum of 10000 trials (the

Table 3: Results of various methods on MIS problems. Notice that we disable graph reduction and 2-opt local search in all models for a fair comparison, since it is pointed out by [8] that all models would perform similarly with a local search post-processing. See Section 4.2.2 for detailed descriptions. * indicates the baseline for computing the performance drop.

| Method | Type | SATLIB | | | ER-[700-800] | | | ER-[9000-11000] | | |
|---|---|---|---|---|---|---|---|---|---|---|
| | | Size ↑ | Drop ↓ | Time ↓ | Size ↑ | Drop ↓ | Time ↓ | Size ↑ | Drop ↓ | Time ↓ |
| KaMIS | OR | 425.96* | — | 37.58m | 44.87* | — | 52.13m | 381.31* | — | 7.6h |
| Gurobi | OR | 425.95 | 0.00% | 26.00m | 41.38 | 7.78% | 50.00m | N/A | N/A | N/A |
| Intel | SL+TS | N/A | N/A | N/A | 38.80 | 13.43% | 20.00m | N/A | N/A | N/A |
| Intel | SL+G | 420.66 | 1.48% | 23.05m | 34.86 | 22.31% | 6.06m | 284.63 | 25.35% | 5.02m |
| DGL | SL+TS | N/A | N/A | N/A | 37.26 | 16.96% | 22.71m | N/A | N/A | N/A |
| LwD | RL+S | 422.22 | 0.88% | 18.83m | 41.17 | 8.25% | 6.33m | **345.88** | **9.29%** | 7.56m |
| DIMES (ours) | RL+G | 421.24 | 1.11% | 24.17m | 38.24 | 14.78% | 6.12m | 320.50 | 15.95% | 5.21m |
| DIMES (ours) | RL+S | **423.28** | **0.63%** | 20.26m | **42.06** | **6.26%** | 12.01m | 332.80 | 12.72% | 12.51m |

default configuration of LKH-3); (ii) *less trials*: we perform 1 run with a maximum of 500 trials for TSP-500/1000 and 250 trials for TSP-10000, so that the running times of LKH-3 match those of DIMES+MCTS. Besides, we also compare DIMES against simple heuristics, including Nearest, Random, and Farthest Insertion.

- Learning-based methods include 8 variants of the 4 methods with the strongest results in recent benchmark evaluations, namely EAN [13], AM [42], GCN [31], POMO+EAS [28], and Att-GCN [19], respectively. Those methods can be further divided into the reinforcement learning (RL) sub-category and the supervised learning (SL) sub-category. Some reinforcement learning methods can further adopt an Active Search (AS) stage to fine-tune on each instance. The results of the baselines except the running time of Att-GCN are taken from Fu et al. [19]. Note that baselines are trained on small graphs and evaluated on large graphs, while DIMES can be trained directly on large graphs. We re-run the publicly available code of Att-GCN on our hardware to ensure fair comparison of time.

The decoding schemes in each method (if applicable) are further specified as Greedy decoding (G), Sampling (S), Beam Search (BS), and Monte Carlo Tree Search (MCTS). The 2-OPT improvements [11] can be optionally used to further improve the neural network-generated solution via heuristic local search. See Section 3.4 for a more detailed descriptions of the various decoding techniques.

As is shown in the table, DIMES significantly outperforms many previous learning-based methods. Notably, although DIMES is trained without any ground truth solutions, it is able to outperform the supervised method. DIMES also consistently outperforms simple traditional heuristics. The best performance is achieved by RL+AS+MCTS, which requires considerably more time. RL+AS+G/S are faster than RL+AS+MCTS and are competitive to the simple heuristics. Removing AS in DIMES shortens the running time and leads to only a slight, acceptable performance drop. Moreover, they are still better than many previous learning-based methods in terms of solution quality and inference time.

### 4.1.3 Ablation Study

**On Meta-Learning** To study the efficacy of meta-learning, we consider two dimensions of ablations: (i) with or without inner gradient updates: whether to use $f_{\boldsymbol{\Phi}}(\boldsymbol{\kappa}_s, \boldsymbol{A}_s)$ or $\boldsymbol{\theta}_s^{(T)}$ in the objective function; (ii) with or without fine-tuning in the inference phase. The results on TSP-1000 with training phase $T = 10$ and greedy decoding are summarized in Table 2a. Both inner updates and fine-tuning are crucial to the performance of our method. That is because meta-learning helps the model generalize across problem instances, and fine-tuning helps the trained model adapt to each specific problem instance.

**On Fine-Tuning Parts** We study the effect of fine-tuning parts during both training and testing. In general, the neural architecture we used is a GNN appended with an MLP, whose output is the continuous parameterization $\boldsymbol{\theta}$. We consider the following fine-tuning parts: (i) the continuous parameterization (Cont. Param.); (ii) the parameter of MLP; (iii) the output of GNN and the parameters of MLP; (iv) the parameters of GNN and MLP. Table 2b summarizes the results with various fine-tuning parts for TSP-1000 with training phase $T = 5$ and greedy decoding. The result

demonstrates that (iii) works best. We conjecture that (iii) makes a nice trade-off between universality and variance reduction.

**On Inner Gradient Update Steps** We also study the effect of the number $T$ of inner gradient update steps during training. Table 2c shows the test performance on TSP-1000 by greedy decoding with various $T$'s. As the number of inner gradient updates increases, the test performance improves accordingly. Meanwhile, more inner gradient update steps consumes more training time. Hence, there is a trade-off between performance and training time in practice.

**On Heatmaps for MCTS** To study where continuous parameterization of DIMES is essential to good performance in MCTS, we replace it with the following heatmaps: (i) each value is independently sampled from $\mathrm{Unif}(0, 1)$; (ii) $1/(r_i + 1)$, where $r_i \geq 1$ denotes the rank of the length of the $i$-th edge among those edges that share the source node with it. This can be regarded as an approximation to the nearest neighbor heuristics. We also compare with the Att-GCN heatmap [19]. Comparison of continuous parameterizations for TSP-1000 by MCTS is shown in Table 2d. The result confirms that the DIMES continuous parameterization does not simply learn nearest neighbor heuristics, but can identify non-trivial good candidate edges.

## 4.2 Experiments For Maximal Independent Set

### 4.2.1 Experimental Settings

**Data Sets** We mainly focus on two types of graphs that recent work [48, 1, 8] shows struggles against, i.e., Erdős-Rényi (ER) graphs [16] and SATLIB [25], where the latter is a set of graphs reduced from SAT instances in CNF. The ER graphs of different scales are named ER-[700-800] and ER-[9000-11000], where ER-[$n$-$N$] indicates the graph contains $n$ to $N$ nodes. The pairwise connection probability $p$ is set to $0.15$ and $0.02$ for ER-[700-800] and ER-[9000-11000], respectively. The 4,096 training and 5,00 test ER graphs are randomly generated. For SATLIB, which consists of 40,000 instances, of which we train on 39,500 and test on 500. Each SAT instance has between 403 to 449 clauses. Since we cannot find the standard train-test splits for both SAT and ER graphs datasets, we randomly split the datasets and re-run all the baseline methods.

**Evaluation Metrics** To compare the solving ability of various methods, we report the average size of the independent set (Size), average performance drop (Drop) and latency time (Time), respectively, where *Size* (the larger, the better) is the average size of the system-predicted maximal independent set for each test-set graph, *Drop* and *Time* are defined similarly as in Section 4.1.1.

**Training and Hardware** Due to the space limit, please refer to the appendix.

### 4.2.2 Main Results

Our main results are summarized in Table 3, where our method (last line) is compared 6 other MIS solvers on the same test sets, including two traditional OR methods (i.e., Gurobi and KaMIS) and four learning-based methods. The active search is not used for MIS evaluation since our preliminary experiments only show insignificant improvements. For Gurobi, we formulate the MIS problem as a integer linear program. For KaMIS, we use the code unmodified from the official repository[5]. The four learning-based methods can be divided into the reinforcement learning (RL) category, i.e., S2V-DQN [36] and LwD [1]) and the supervised learning (SL) category, i.e., Intel [48] and DGL [8].

We produced the results for all the learning-based methods by running an integrated implementation[6] provided by Böther et al. [8]. Notice that as pointed out by Böther et al. [8], the graph reduction and local 2-opt search [3] can find near-optimal solutions even starting from a randomly generated solution, so we disable the local search or graph reduction techniques during the evaluation for all learning based methods to reveal their real CO-solving ability. The methods that cannot produce results in the $10\times$ time limit of DIMES are labeled as N/A.

As is shown in Table 3, our DIMES model outperforms previous baseline methods on the medium-scale SATLIB and ER-[700-800] datasets, and significantly outperforms the supervised baseline (i.e.,

---

[5]`https://github.com/KarlsruheMIS/KaMIS` (MIT License)

[6]`https://github.com/MaxiBoether/mis-benchmark-framework` (No License)

Intel) on the large-scale ER-[9000-11000] setting. This shows that supervised neural CO solvers cannot well solve large-scale CO problem due to the expensive annotation problem and generalization problem. In contrast, reinforcement-learning methods are a better choice for large-scale CO problems. We also find that LwD outperforms DIMES on the large-scale ER-[9000-11000] setting. We believe this is because LwD is specially designed for locally decomposable CO problems such as MIS and thus can use parallel prediction, but DIMES are designed for general CO problems and only uses autoregressive factorization. How to better utilize the fact of local decomposability of MIS-like problems is one of our future work.

## 5 Conclusion & Discussion

Scalability without significantly scarifying the approximation accuracy is a critical challenge in combinatorial optimization. In this work we proposed DIMES, a differentiable meta solver that is able to solve large-scale combinatorial optimization problems effectively and efficiently, including TSP and MIS. The novel parts of DIMES include the compact continuous parameterization and the meta-learning strategy. Notably, although our method is trained without any ground truth solutions, it is able to outperform several supervised methods. In comparison with other strong DRL solvers on TSP and MIS problems, DIMES can scale up to graphs with ten thousand nodes while the others either fail to scale up, or can only produce significantly worse solutions instead in most cases.

Our unified framework is not limited to TSP and MIS. Its generality is based on the assumption that each feasible solution of the CO problem on hand can be represented with 0/1 valued variables (typically corresponding the selection of a subset of nodes or edges), which is fairly mild and generally applicable to many CO problems beyond TSP and MIS (see Karp's 21 NP-complete problems [35]) with few modifications. The design principle of auxiliary distributions is to design an autoregressive model that can sequentially grow a valid partial solution toward a valid complete solution. This design principle is also proven to be general enough for many problems in neural learning, including CO solvers. There do exist problems beyond this assumption, e.g., Mixed Integer Programming (MIP), where variables can take multiple integer values instead of binary values. Nevertheless, Nair et al. [56] showed that this issue can be addressed by reducing each integer value within range $[U]$ to a sequence of $\lceil \log_2 U \rceil$ bits and by predicting the bits from the most to the least significant bits. In this way, a multi-valued MIP problem can be reduced to a binary-valued MIP problem with more variables.

One limitation of DIMES is that the continuous parameterization $\boldsymbol{\theta}$ is generated in one-shot without intermediate steps, which could potentially limit the reasoning power of our method, as is shown in the MIS task. Another limitation is that applying DIMES to a broader ranges of NP-complete problems that variables can take multiple values, such as Mixed Integer Programming (MIP), is non-trivial and needs further understanding of the nature of the problems.

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
