# A  Additional Related Work

## A.1  Per-Instance Search

Once the neural network is trained over a collection of problem instances, per-instance fine-tuning can be used to improve the quality of solutions via local search. For DRL solvers, Bello et al. [7] fine-tuned the policy network on each test graph, which is referred as *active search*. Hottung et al. [28] proposed three active search strategies for efficient updating of parameter subsets during search. Zheng et al. [77] tried a combination of traditional reinforcement learning with Lin-Kernighan-Helsgaun (LKH) Algorithm [49, 24]. Hottung et al. [27] performed per-instance search in a differentiable continuous space encoded by a conditional variational auto-encoder [39]. With a heatmap indicating the promising parts of the search space, discrete solutions can be found via beam search [31], sampling [42], guided tree-search [48], dynamic programming [44], and Monte Carlo Tree Search (MCTS) [19]. In this paper, we mainly adopt greedy, sampling, and MCTS as the per-instance search techniques.

# B  Per-instance Search

In this section, we describe the decoding strategies used in our paper. Given a fine-tuned (i.e., after active search) continuous parameterization $\boldsymbol{\theta}_s^{(T)}$ of the solution space, the per-instance search decoding aims to search for a feasible solution that minimizes the cost function $c_s$.

**Greedy Decoding**    generates the solution through a sequential decoding process similar to the auxiliary distribution designed for each combinatorial optimization problem, where at each step, the variable $k$ with the highest score $\theta_k$ is chosen to extend the partial solution. For TSP, the first node in the permutation is picked at random.

**Sampling**    Inspired by Kool et al. [42], we propose to parallelly sample multiple solutions according to the auxiliary distribution and report the best one. The continuous parameterization is divided by a temperature parameter $\tau$. The parallel sampling of solutions in DIMES is very efficient due to the fact that it only relies on the final parameterization $\boldsymbol{\theta}_s^{(T)}/\tau$ but not on neural networks.

**Monte Carlo Tree Search**    Inspired by [19], for the TSP task, we also leverage a more advanced reinforcement learning-based searching approach, i.e., Monte Carlo tree search (MCTS), to find high-quality solutions. In MCTS, $k$-opt transformation actions are sampled guided by the continuous parameterization $\boldsymbol{\theta}_s^{(T)}$ to improve the current solutions. The MCTS iterates over the simulation, selection, and back-propagation steps, until no improving actions exists among the sampling pool. For more details, please refer to [19].

# C  Implementation Details

## C.1  Neural Architecture for TSP

**Anisotropic Graph Neural Networks**    We follow Joshi et al. [33] on the choice of neural architectures. The backbone of the graph neural network is an anisotropic GNN with an edge gating mechanism [9]. Let $\boldsymbol{h}_i^\ell$ and $\boldsymbol{e}_{ij}^\ell$ denote the node and edge features at layer $\ell$ associated with node $i$ and edge $ij$, respectively. The features at the next layer is propagated with an anisotropic message passing scheme:

$$\boldsymbol{h}_i^{\ell+1} = \boldsymbol{h}_i^\ell + \alpha(\text{BN}(\boldsymbol{U}^\ell \boldsymbol{h}_i^\ell + \mathcal{A}_{j \in \mathcal{N}_i}(\sigma(\boldsymbol{e}_{ij}^\ell) \odot \boldsymbol{V}^\ell \boldsymbol{h}_j^\ell))), \tag{18}$$

$$\boldsymbol{e}_{ij}^{\ell+1} = \boldsymbol{e}_{ij}^\ell + \alpha(\text{BN}(\boldsymbol{P}^\ell \boldsymbol{e}_{ij}^\ell + \boldsymbol{Q}^\ell \boldsymbol{h}_i^\ell + \boldsymbol{R}^\ell \boldsymbol{h}_j^\ell)). \tag{19}$$

where $\boldsymbol{U}^\ell, \boldsymbol{V}^\ell, \boldsymbol{P}^\ell, \boldsymbol{Q}^\ell, \boldsymbol{R}^\ell \in \mathbb{R}^{d \times d}$ are the learnable parameters of layer $\ell$, $\alpha$ denotes the activation function (we use SiLU [15] in this paper), BN denotes the Batch Normalization operator [30], $\mathcal{A}$ denotes the aggregation function (we use mean pooling in this paper), $\sigma$ is the sigmoid function, $\odot$ is the Hadamard product, and $\mathcal{N}_i$ denotes the outlinks (neighborhood) of node $i$. We use a 12-layer GNN with width 32.

The node and edge features at the first layer $\boldsymbol{h}_i^0$ and $\boldsymbol{e}_{ij}^0$ are initialized with the absolute position of the nodes and absolute length of the edges, respectively. After the anisotropic GNN backbone, a Multi-Layer Perceptron (MLP) is appended and generates the final continuous parameterization $\boldsymbol{\theta}$ for all the edges. We use a 3-layer MLP with width 32.

**Graph Sparsification**   As described, we focus on developing a neural TSP solver for graphs with tens of thousands of nodes. Because the number of edges in the graph grows quadratically to the number of nodes, a densely connected graph is intractable for an anisotropic GNN when it is applied to large graphs. Therefore, we use a simple heuristic to sparsify the original graph. Specifically, we prune the outlinks of each node such that it is only connected to $k$ nearest neighbors. The continuous parameterization $\boldsymbol{\theta}$ is also pruned accordingly. As a result, the computation complexity of our method is reduced from $O(n^2)$ to $O(nk)$, where $n$ is the number of nodes in the graph.

## C.2   Neural Architecture for MIS

**Graph Convolutional Networks**   We follow Li et al. [48] on the choice of neural architecture, i.e., using Graph Convectional Network (GCN) [40], since $\boldsymbol{\theta}$ is merely scores for each node. Specifically, the GCN backbone consists of multiple layers $\{\mathbf{h}^l\}$ where $\mathbf{h}^l \in \mathbb{R}^{N \times C^l}$ is the feature layer in the $l$-th layer and $C^l$ is the number of feature channels in the $l$-th layer. We initialize the input layer $\mathbf{h}^0$ with all ones and $\mathbf{h}^{l+1}$ is computed from the previous layer $\mathbf{h}^l$ with layer-wise convolutions:

$$\mathbf{h}^{l+1} = \sigma(\mathbf{h}^l \mathbf{U}_0^l + \mathbf{D}^{-\frac{1}{2}} \mathbf{A} \mathbf{D}^{-\frac{1}{2}} \mathbf{h}^l \mathbf{U}_1^l), \tag{20}$$

where $\mathbf{U}_0^l \in \mathbb{R}^{C^l \times C^{l+1}}$ and $\mathbf{U}_1^l \in \mathbb{R}^{C^l \times C^{l+1}}$ are trainable weights in the convolutions of the network, $\mathbf{D}$ is the degree matrix of $\mathbf{A}$ with its diagonal entry $\mathbf{D}(i, i) = \sum_j \mathbf{A}(j, i)$, and $\sigma(\cdot)$ is the ReLU [55] activation function. After the GCN backbone, a 10-layer Multi-Layer Perceptron (MLP) with residual connections [22] is appended and generates the final continuous parameterization $\boldsymbol{\theta}$ for all the nodes.

# D   Experimental Details

## D.1   TSP

**Training**   For TSP-500, we train our model for 120 meta-gradient descent steps (1.5 h in total) with $T = 15$. For TSP-1000, we train our model for 120 meta-gradient descent steps (1.7 h in total) with $T = 14$. For TSP-10000, we train our model for 50 meta-gradient descent steps (10 h in total) with $T = 12$. We generate 3 instances per meta-gradient descent step. We use the AdamW optimizer [50] with learning rate 0.005 and weight decay 0.0005 for meta-gradient descent steps, and with learning rate 0.05 for REINFORCE gradient descent steps. For other learning-based baseline methods, we download and rerun the source codes published by their original authors based on their pre-trained models.

**Hardware**   We follow the hardware environment suggested by Fu et al. [19]. For the three traditional algorithms, since their source codes do not support running on GPUs, they run on Intel Xeon Gold 5118 CPU @ 2.30GHz. To ensure fair comparison, learning-based methods run on GTX 1080 Ti GPU during the testing phase. MCTS runs on Intel Xeon Gold 6230 80-core CPU @ 2.10GHz, where we use 64 threads for TSP-500 and TSP-1000, and 16 threads for TSP-10000. For the training phase, we train our model on NVIDIA Tesla P100 16GB GPU.

**Reproduction**   We implement DIMES for TSP based on PyTorch Geometric [17] in LibTorch and PyTorch [62]. Our code for TSP is publicly available.[7] The test instances are provided by Fu et al. [19].[8]

## D.2   MIS

**Training**   For SAT, we train our model for 50k meta-gradient steps with $T = 1$. For ER-[700-800], we train our model for 150k meta-gradient steps with $T = 1$. For ER-[9000-11000], we initialize

---

[7]`https://github.com/DIMESTeam/DIMES` (MIT license)
[8]`https://github.com/Spider-scnu/TSP` (MIT license)

our model from the checkpoint of ER-[700-800], and further train it for 200 meta-gradient steps. We use a batch size of 8 on all datasets and Adam optimizer [38] with learning rate 0.001 for the meta-gradient descent step, and with learning rate 0.0002 for REINFORCE gradient descent steps. For other learning-based baseline methods, we mainly use an integrated implementation[9] provided by Böther et al. [8].

**Hardware**    All the methods are trained and evaluated on a single NVIDIA Ampere A100 40 GB GPU, with AMD EPYC 7713 64-Core CPUs.

**Reproduction**    Our code for MIS is publicly available.[10] Following Böther et al. [8], for SAT, we use the "Random-3-SAT Instances with Controlled Backbone Size" dataset[11] and randomly split it into 39500 training instances and 500 test instances. For the Erdős-Rényi graphs, both training and test instances are randomly generated.

# E    Proofs

In this section, we follow the notation introduced in Section 3.

## E.1    Convergence of Solution Distributions

The following propositions show that $p_{\boldsymbol{\theta}}$ and $q_{\boldsymbol{\theta}}$ converge to the *same* solution. They imply that we can optimize $q_{\boldsymbol{\theta}}$ instead of $p_{\boldsymbol{\theta}}$.

**Proposition 1** (TSP version). *Let $0 < \delta \ll 1$ be a sufficiently small number. If $q_{\boldsymbol{\theta}}^{\text{TSP}}(f) \geq 1 - \delta$ for a solution $f \in \mathcal{F}$, then we also have $p_{\boldsymbol{\theta}}(f) \geq 1 - O(\delta)$.*

**Proposition 2** (MIS version). *Suppose that $\boldsymbol{\theta}$ is normalized (i.e., $\sum_i \exp(\theta_i) = 1$) and uniformly bounded w.r.t. a solution $f \in \mathcal{F}$ (i.e., $\sum_i f_i \exp(\theta_i) / \exp(\sum_i f_i \theta_i) \leq L$ for a constant $L > 0$). Let $0 < \delta \ll 1$ be a sufficiently small number. If $q_{\boldsymbol{\theta}}^{\text{MIS}}(f) \geq 1 - \delta$, then we also have $p_{\boldsymbol{\theta}}(f) \geq 1 - O(\delta)$.*

*Remark.* Propositions 1 & 2 imply that if $q_{\boldsymbol{\theta}}$ converges to $f$ ($\delta \to 0_+$), then $p_{\boldsymbol{\theta}}$ also converges to $f$.

*Proof for TSP.* Using the bound of $q_{\boldsymbol{\theta}}^{\text{TSP}}(f)$, we have for any node $j$:

$$q_{\text{TSP}}(\pi_f \mid \pi_f(0) = j) = n q_{\boldsymbol{\theta}}^{\text{TSP}}(f) - \sum_{i \neq j} q_{\text{TSP}}(\pi_f \mid \pi_f(0) = i) \tag{21}$$

$$\geq n q_{\boldsymbol{\theta}}^{\text{TSP}}(f) - (n - 1) \tag{22}$$

$$\geq n(1 - \delta) - (n - 1) = 1 - O(\delta). \tag{23}$$

Thus, for any edge $(i, j)$ in the tour $\pi_f$ and any edge $(i, k) \neq (i, j)$,

$$\theta_{i,j} - \theta_{i,k} = \log \frac{\exp(\theta_{i,j})}{\exp(\theta_{i,k})} \tag{24}$$

$$\geq \log \frac{q_{\boldsymbol{\theta}}(\pi_f(1) = j \mid \pi_f(0) = i)}{1 - q_{\boldsymbol{\theta}}(\pi_f(1) = j \mid \pi_f(0) = i)} \tag{25}$$

$$\geq \log \frac{q_{\boldsymbol{\theta}}(\pi_f \mid \pi_f(0) = i)}{1 - q_{\boldsymbol{\theta}}(\pi_f \mid \pi_f(0) = i)} \tag{26}$$

$$\geq \log \frac{1 - O(\delta)}{O(\delta)}. \tag{27}$$

Note that for any edge $(i, j)$ in the tour $f$ (denoted by $(i, j) \in \pi_f$) and any solution $g \in \mathcal{F} \setminus \{f\}$, there exist a unique $k_i^g$ such that edge $(i, k_i^g)$ is in the tour $\pi_g$, and $(i, k_i^g) \neq (i, j)$ for at least one

---

[9]`https://github.com/MaxiBoether/mis-benchmark-framework` (No license)

[10]`https://github.com/DIMESTeam/DIMES` (MIT license)

[11]`https://www.cs.ubc.ca/~hoos/SATLIB/Benchmarks/SAT/CBS/descr_CBS.html`

edge $(i,j) \in \pi_f$. Then,

$$p_{\boldsymbol{\theta}}(f) = \frac{1}{1 + \sum_{g \in \mathcal{F} \backslash \{f\}} \exp\left(-\sum_{(i,j) \in f} (\theta_{i,j} - \theta_{i,k_i^g})\right)} \tag{28}$$

$$= \frac{1}{1 + \sum_{g \in \mathcal{F} \backslash \{f\}} \exp\left(-\sum_{(i,j) \in f \backslash g} (\theta_{i,j} - \theta_{i,k_i^g})\right)} \tag{29}$$

$$\geq \frac{1}{1 + \sum_{g \in \mathcal{F} \backslash \{f\}} \exp\left(-\sum_{(i,j) \in f \backslash g} \log \frac{1 - O(\delta)}{O(\delta)}\right)} \tag{30}$$

$$= 1 - O(\delta). \tag{31}$$

$$\square$$

*Proof for MIS.* Let $|g|$ denote the size of a solution $g \in \mathcal{F}$, i.e., $|g| = \sum_i g_i$. With a little abuse of notation, let $g \in \mathcal{F}$ also denote the corresponding independent set. Note that

$$\frac{\max_{i \notin f} \exp(\theta_i)}{\max_{i \notin f} \exp(\theta_i) + \sum_{i \in f} \exp(\theta_i)} \tag{32}$$

$$\leq \frac{\sum_{i \notin f} \exp(\theta_i)}{\sum_{i \notin f} \exp(\theta_i) + \sum_{i \in f} \exp(\theta_i)} \tag{33}$$

$$= \frac{\sum_{i \notin f} \exp(\theta_i)}{\sum_i \exp(\theta_i)} = \sum_{i \notin f} q_{\mathrm{MIS}}(a_1 = i) \tag{34}$$

$$= q_{\mathrm{MIS}}(a_1 \notin f) \leq 1 - q_{\boldsymbol{\theta}}^{\mathrm{MIS}}(f) \leq \delta. \tag{35}$$

This implies

$$\max_{i \notin f} \exp(\theta_i) \leq \frac{\delta}{1 - \delta} \sum_{i \in f} \exp(\theta_i). \tag{36}$$

Recall that we have assumed in Section 3.2.2 that each $f' \in \mathcal{F}$ is not a proper subset of any other $f'' \in \mathcal{F}$. Thus for any $f, g \in \mathcal{F}$, we have $f \backslash g \neq \varnothing$, and $g \backslash f \neq \varnothing$. Note also that $\exp(\theta_i) \leq \sum_j \exp(\theta_j) = 1$ for all nodes $i$. Hence,

$$p_{\boldsymbol{\theta}}(f) = \left(1 + \sum_{g \in \mathcal{F} \backslash \{f\}} \frac{\exp(\sum_i g_i \theta_i)}{\exp(\sum_i f_i \theta_i)}\right)^{-1} \tag{37}$$

$$= \left(1 + \sum_{g \in \mathcal{F} \backslash \{f\}} \frac{\prod_{i \in g \backslash f} \exp(\theta_i)}{\prod_{i \in f \backslash g} \exp(\theta_i)}\right)^{-1} \tag{38}$$

$$\geq \left(1 + \sum_{g \in \mathcal{F} \backslash \{f\}} \frac{\max_{i \in g \backslash f} \exp(\theta_i)}{\prod_{i \in f \backslash g} \exp(\theta_i)}\right)^{-1} \tag{39}$$

$$\geq \left(1 + \sum_{g \in \mathcal{F} \backslash \{f\}} \frac{\max_{i \notin f} \exp(\theta_i)}{\prod_{i \in f} \exp(\theta_i)}\right)^{-1} \tag{40}$$

$$\geq \left(1 + \sum_{g \in \mathcal{F} \backslash \{f\}} \frac{\frac{\delta}{1-\delta} \sum_{i \in f} \exp(\theta_i)}{\prod_{i \in f} \exp(\theta_i)}\right)^{-1} \tag{41}$$

$$\geq \left(1 + \sum_{g \in \mathcal{F} \backslash \{f\}} \frac{\delta}{1 - \delta} \cdot L\right)^{-1} \tag{42}$$

$$= 1 - O(\delta). \tag{43}$$

$$\square$$

### E.2 First-Order Approximation of Meta-Gradient

The following proposition gives a first-order approximation formula of the meta-gradient.

**Proposition 3.** *Let $F_{\boldsymbol{\Phi}}(\kappa_s, A_s)$ be a GNN $F$ with parameter $\boldsymbol{\Phi}$ and input $(\kappa_s, A_s)$, $\mathcal{L}(\boldsymbol{\Phi} \mid \{s\})$ be a loss function, and $\alpha > 0$ be a learning rate. Suppose $\boldsymbol{\Phi}_s^{(0)} = \boldsymbol{\Phi}$, and $\boldsymbol{\Phi}_s^{(t)} = \boldsymbol{\Phi}_s^{(t-1)} - \alpha \nabla_{\boldsymbol{\Phi}_s^{(t-1)}} \mathcal{L}(\boldsymbol{\Phi}_s^{(t-1)} \mid \{s\})$ for $1 \leq t \leq T$, and $\boldsymbol{\theta}_s^{(T)} = F_{\boldsymbol{\Phi}_s^{(T)}}(\kappa_s, A_s)$. Then,*

$$\nabla_{\boldsymbol{\Phi}} \boldsymbol{\theta}_s^{(T)} = \nabla_{\boldsymbol{\Phi}_s^{(T)}} F_{\boldsymbol{\Phi}_s^{(T)}}(\kappa_s, A_s) + O(\alpha).$$

*Proof.* The proof resembles [58]. By chain rule,

$$\nabla_{\boldsymbol{\Phi}_s^{(0)}} \boldsymbol{\Phi}_s^{(T)} = \prod_{t=1}^{T} \nabla_{\boldsymbol{\Phi}_s^{(t-1)}} \boldsymbol{\Phi}_s^{(t)} \tag{44}$$

$$= \prod_{t=1}^{T} \nabla_{\boldsymbol{\Phi}_s^{(t-1)}} (\boldsymbol{\Phi}_s^{(t-1)} - \alpha \nabla_{\boldsymbol{\Phi}_s^{(t-1)}} \mathcal{L}(\boldsymbol{\Phi}_s^{(t-1)} \mid \{s\})) \tag{45}$$

$$= \prod_{t=1}^{T} (\boldsymbol{I} - \alpha \nabla_{\boldsymbol{\Phi}_s^{(t-1)}}^2 \mathcal{L}(\boldsymbol{\Phi}_s^{(t-1)} \mid \{s\})) \tag{46}$$

$$= \boldsymbol{I} + \sum_{k=1}^{T} (-\alpha)^k \sum_{1 \leq t_1 < \cdots < t_k \leq T} \prod_{i=1}^{k} \nabla_{\boldsymbol{\Phi}_s^{(t_i-1)}}^2 \mathcal{L}(\boldsymbol{\Phi}_s^{(t_i-1)} \mid \{s\}) \tag{47}$$

$$= \boldsymbol{I} + O(\alpha). \tag{48}$$

Hence,

$$\nabla_{\boldsymbol{\Phi}} \boldsymbol{\theta}_s^{(T)} = \nabla_{\boldsymbol{\Phi}_s^{(0)}} \boldsymbol{\Phi}_s^{(T)} \nabla_{\boldsymbol{\Phi}_s^{(T)}} F_{\boldsymbol{\Phi}_s^{(T)}}(\kappa_s, A_s) \tag{49}$$

$$= (\boldsymbol{I} + O(\alpha)) \nabla_{\boldsymbol{\Phi}_s^{(T)}} F_{\boldsymbol{\Phi}_s^{(T)}}(\kappa_s, A_s) \tag{50}$$

$$= \nabla_{\boldsymbol{\Phi}_s^{(T)}} F_{\boldsymbol{\Phi}_s^{(T)}}(\kappa_s, A_s) + O(\alpha). \tag{51}$$

$\square$

## F   Additional Experiments for TSP

### F.1   Performance on TSP-100

We trained DIMES on TSP-100 and evaluate it on TSP-100 with $T = 10$ and 0 (i.e., with and without meta-learning). Since MCTS is the best per-instance search scheme for DIMES (see Table 1), we also use MCTS here. When using AS, we fine-tune DIMES on each instance for 100 steps. We compare DIMES with learning-based methods listed in Section 4.1.2. Results of baselines are taken from Fu et al. [19]. The results are presented in Table 4.

As is shown in the table, DIMES outperforms all learning-based methods, and its results are very close to optimal lengths given by exact solvers. The results suggest that DIMES achieves the best in-distribution performance among learning-based methods. Notably, with meta-learning ($T = 10$), even when DIMES does not fine-tune (i.e., no active search) for each problem instance in evaluation, it still outperforms all other learning-based methods. This again demonstrates the efficacy of meta-learning to DIMES.

### F.2   Extrapolation Performance

We evaluate the exptrapolation performance of DIMES (i.e., trained on smaller graphs and tested on larger graphs). We train the model on TSP-100 and test it on TSP-500/1000/10000. For testing, we use RL+S ($\tau = 0.01$) without active search. The results are reported in Table 5 in comparison with corresponding results trained on larger graphs (TSP-$n$).

Table 4: Results on TSP-100. * indicates the baseline for computing the performance drop.

| Method | Type | Length ↓ | Drop ↓ |
|--------|------|----------|--------|
| Concorde | OR (exact) | 7.7609* | — |
| Gurobi | OR (exact) | 7.7609* | — |
| LKH-3 | OR | 7.7611 | 0.0026% |
| EAN | RL+S | 8.8372 | 13.8679% |
| EAN | RL+S+2-OPT | 8.2449 | 6.2365% |
| AM | RL+S | 7.9735 | 2.7391% |
| AM | RL+G | 8.1008 | 4.3791% |
| AM | RL+BS | 7.9536 | 2.4829% |
| GCN | SL+G | 8.4128 | 8.3995% |
| GCN | SL+BS | 7.8763 | 1.4828% |
| Att-GCN | SL+MCTS | 7.7638 | 0.0370% |
| DIMES ($T = 0$) | RL+MCTS | 7.7647 | 0.0490% |
| DIMES ($T = 0$) | RL+AS+MCTS | 7.7618 | 0.0116% |
| DIMES ($T = 10$) | RL+MCTS | 7.7620 | 0.0142% |
| DIMES ($T = 10$) | RL+AS+MCTS | **7.7617** | **0.0103%** |

Table 5: Results of DIMES (RL+S). "Trained on TSP-100" indicates extrapolation performance.

| Setting | TSP-500 | | TSP-1000 | | TSP-10000 | |
|---------|---------|--------|----------|--------|-----------|--------|
| | Length ↓ | Drop ↓ | Length ↓ | Drop ↓ | Length ↓ | Drop ↓ |
| Trained on TSP-$n$ | 18.84 | 13.84% | 26.36 | 14.01% | 85.75 | 19.48% |
| Trained on TSP-100 | 19.21 | 16.07% | 27.21 | 17.69% | 86.24 | 20.16% |

From the table we can observe that the performance of DIMES does not drop much, which demonstrates the nice extrapolation performance of DIMES. One of our hypotheses is that graph sparsification in our neural network (see Appendix C.1) avoids the explosion of activation values in the graph neural network. Another hypothesis is that meta learning tends to not generate too extreme values in (see point 9 of our previous response) and hence improve the generalization capability.

### F.3 Stability of Training

We compare the training settings of AM [42], POMO [45], and DIMES in Table 6. The training costs of AM and POMO are obtained from their papers[12] A training step means a gradient descent step of the GNN. That is, for AM/POMO, a training step means a gradient descent step over a batch; for DIMES, a training step means a meta-gradient descent step.

The table shows that DIMES is much more sample-efficient than AM/POMO. Notably, DIMES achieves stable training using only 3 instances per meta-gradient descent step. Hence, its total training time is accordingly much shorter, even though its per-step time is longer. Moreover, the stability of training enables us to use a larger learning rate, which also accelerates training.

To further illustrate the fast stable training of DIMES, we compare the dynamics of training among AM, POMO, and DIMES in Figure 1. We closely follow the training settings of their papers, i.e., we train AM/POMO on TSP-100 and DIMES on TSP-500. For AM/POMO, we train their models on our hardware by re-running their public source code. The performance is evaluated using TSP-500 test instances. For DIMES, we use RL+S in evaluation.

From Figure 1a, we can observe that DIMES stably converges to a better performance within fewer time, while the dynamics of training AM/POMO are slower and less stable. From Figure 1b, we can observe that DIMES converges at much fewer training steps. The results again demonstrate that the training of DIMES is fast and stable.

---

[12]For AM/POMO, per-step training time is estimated by total training time divided by total training steps.

Table 6: Comparison of training settings for TSP-500/1000/10000.

| Setting | AM | POMO | DIMES |
|---|---|---|---|
| Training problem scale | TSP-100 | TSP-100 | TSP-500 / 1000 / 10000 |
| Training descent steps | 250,000 | 312,600 | 120 / 120 / 50 |
| Per-step training instances | 512 | 64 | 3 |
| Total training instances | 128,000,000 | 20,000,000 | 360 / 360 / 150 |
| Per-step training time | 0.66 s | 0.28 s | 45 s / 51 s / 12 m |
| Total training time | 2 d | 1 d | 1.5 h / 1.7 h / 10 h |
| Training GPUs | 2 | 1 | 1 |

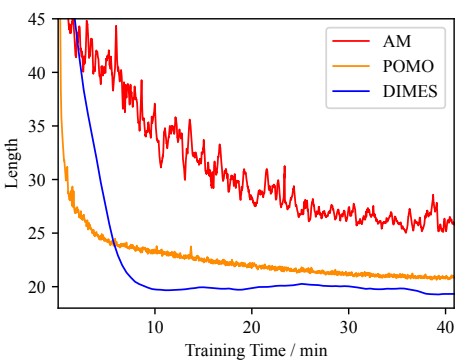

(a) Performance vs training time.

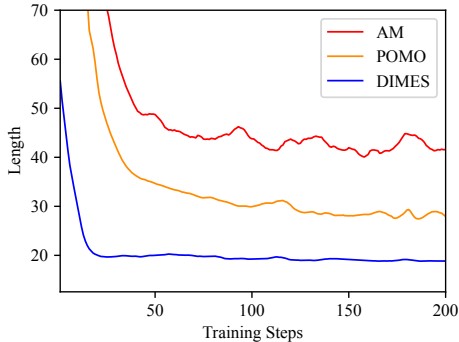

(b) Performance vs training steps.

Figure 1: Evaluation performance vs training cost.