# OpenReview forum: "DIMES: A Differentiable Meta Solver for Combinatorial Optimization Problems"
_NeurIPS.cc/2022/Conference — NeurIPS 2022 Accept_

### Official Review · Reviewer_fe3B · 2022-07-08

**Rating:** 5
**Confidence:** 4
**Soundness:** 3 good
**Presentation:** 3 good
**Contribution:** 2 fair

**Summary:**

The paper proposed DIMES, a framework for tackling graph-based combinatorial optimisation problems with reinforcement learning.  DIMES runs a graph neural network to produce a heatmap over the candidate variables (edges for TSP, nodes for MIS) which corresponds to a policy for which variable to select next from any given position.  This approach seeks to provide more scalable CO solvers as solutions can be efficiently sampled from a generated heatmap with minimal additional overhead.  Compared to prior works, DIMES primary claims of novelty are (i) the process does not require supervised data to generate the heatmaps and (ii) that DIMES can be trained, using meta-learning, on a distribution of instances such that it can quickly and efficiently specialise during active search on a singe instance.  In experiments on TSP and MIS, DIMES is shown to perform competitively or better than the considered RL/SL baselines and remain efficient even on large problem sizes.


**Questions:**

- Regarding the fine-tuning of DIMES in table 1: (i) Is the fine-tuning time included in the run time (I would have expected so, but then I am surprised how, for example, DIMES is faster the AM with equivalent settings if it has to be fine-tuned first)? (ii) Which elements of the model are fine-tuned (as four variants are show in Table 2(b) it is clear there is some choice to be made)?
- If all baselines were taken from Fu et al, can you comment on whether the training of these was a fair comparison to DIMES?  For example, DIMES is trained on the target problem sizes but it is common to present results on larger TSP instances of models trained on smaller problems.
- In the meta-learning ablations, why would training the network using meta-learning (targeted to give good performance after T steps of fine-tuning on a specific instance), give better performance when T=10 during training and T=0 at inference (i.e why is row 2 of Table 2a better than row 1?)
- Why is S2V-DQN not used as a baseline in Table 3, when it is stated to be a baseline on line 297?

**Limitations:**

Yes.

**Strengths And Weaknesses:**

*Strengths*

The problem tacked — scalable RL approaches for CO — is significant and well motivated.  Improvements in performance and scalability of are of interest to the community.

As it is typically intractable to produce optimal solutions to NP-hard CO problems, the efficient sampling of candidate solutions and active search on a target instance at test time important components of many approaches.  In this context, the demonstrated efficacy meta-learning good starting points for the agent from which to fine-tune is interesting.  Whilst it was also novel to my knowledge, a quick search did show that the authors might want to consider softening their claim (line 51) that "we are the first to apply meta-learning over a collection of CO problem instances” (e.g. https://arxiv.org/abs/2105.02741).

The experimental results appear impressive, though I have some points of concern or that require clarification (see below) before I can fairly judge these.

*Weaknesses*

I am not convinced by the authors claim that DIMES “introduces a compact continuous space for parameterizing the underlying distribution of candidate solutions” is novel or significant, nor that this new approach “addresses the scalability challenge in large-scale combinatorial optimization” more so than previous works.  Ultimately, DIMES is predicting a heatmap with a single pre-processing step of the entire problem instance (graph), which then guides some search process (greedy, sampling, MCTS etc).  Whilst the paper contrasts DIMES to previous works with “we introduce a compact continuous space for parameterizing the underlying distribution of candidate solutions” — it seems to me that this is just different language for talking about a heatmap which, by the authors admission, many previous works have already used.  The authors note that previous heatmap approaches were not pure RL (relying on some labelled data), however applying RL for TSP well established and having that agent output a heatmap instead of, say, embeddings that implicitly define a heatmap via attention, or running a full inference at each step is a straightforward modification that doesn’t appear to rely on any architectural or theoretical insights.

Whilst the paper is well written from a linguistic perspective, I do feel it loses clarity in the way it presents DIMES and draws contrasts with prior work where they may not be justified.  For example, “Instead of previous DIR-based CO sovles which rely on construction or improvement heuristics, we introduce a compact continuous space for parameterizing the underlying distribution of candidate solutions which allows massively parallel on-policy sampling”.  To me, DIMES is a construction heuristic (it starts at a node/edge, and then iteratively picks the next until a solution is found).  Moreover, the scalability does not come from any breakthrough regarding the compact continuous parameterisation space, but in essence because as much processing as possible (e.g. the expensive GNN) is restricted to single preprocessing step.  However, this is not a novel idea - even the authors own RL baselines for TSP - Joshi et al [24] and Kool et al [34] - use an expensive preprocessing step followed by fast action decoding.  In this vein, another work the authors may consider relevant is that of Driori et al (https://arxiv.org/pdf/2006.03750) who use dot-product attention between graph embeddings as an alternative parameterisation of a heatmap and demonstrate impressive performance on TSP (outperforming Farthest/2-opt heuristics on most “real world” graphs of up to ~125k nodes).

I am concerned that certain more recent baselines may be missing, and that certain results may be missing from the experimental section.  Some points are listed here, but others are left as questions below where I feel I need more clarity.

- The attention model of Kool et al [34] is used as an RL baseline.  However, this approach has been significantly improved in POMO (https://arxiv.org/abs/2010.16011) and then subsequently adapted for active search at test time (https://arxiv.org/abs/2106.05126).  Driori et al (above) is also a significantly scalable RL TSP solver that would seem to be an important baseline if DIMES is justify claims that they are contributing with regards to scalability.

- The authors only rerun one of the presented baselines (Att-GCN) on their hardware, so timing results for other baselines taken from Fu et al are potentially misleading.

- In Table 3, why is LwD with sampling (RL+S) not included?  As LwD is better than DIMES in greedy mode, but DIMES is better with sampling, would LwD with sampling not be expected to be best of all?

---

> ### Author Response · Authors · 2022-08-02
> **Response (I)**
>
> We thank the reviewer for their time, insightful comments, and questions. We have provided our responses below.
>
> > 1. “I am not convinced by the authors claim that DIMES “introduces a compact continuous space for parameterizing the underlying distribution of candidate solutions” is novel or significant, nor that this new approach “addresses the scalability challenge in large-scale combinatorial optimization” more so than previous works.”
>
> The heatmaps in previous works are generated in a fully non-autoregressive manner, where the values in the heatmap correspond to the estimated probability of each edge to be included in the optimal solution. In our work, the n-by-n matrix $\theta$ is used to parameterize distribution $q$, in an autoregressive manner,  in estimating the probability of the next node conditioned on the path (partial solution) so far. That is, the elements in our $\theta$ matrix are not the estimated probabilities of edges. In other words, although the heatmaps in previous work look like our $\theta$ syntactically (as both are in the form of an n-by-n matrix), the semantics behind them are fundamentally different. To avoid possible confusion, we have revised our paper to make the distinction between “heatmap” in previous work and our novel approach clearer, by calling our $\theta$ matrix “continuous parameterization” instead of “heatmap”.
>
> > 2. “Whilst it was also novel to my knowledge, a quick search did show that the authors might want to consider softening their claim (line 51) that "we are the first to apply meta-learning over a collection of CO problem instances” (e.g. https://arxiv.org/abs/2105.02741).”
>
> We thank the reviewer for the comment. Notice that Zhang et al. applies meta learning to multiobjective optimization where each objective is treated as an optimization task. However, what we wrote in the paper is “To our knowledge, we are the first to apply meta-learning over a collection of CO problem instances, where each instance graph is treated as one of a collection tasks a sub-task in a unified framework.” We believe such a claim is still valid.
>
> > 3. “In this vein, another work the authors may consider relevant is that of Driori et al (https://arxiv.org/pdf/2006.03750) who use dot-product attention between graph embeddings as an alternative parameterisation of a heatmap and demonstrate impressive performance on TSP (outperforming Farthest/2-opt heuristics on most “real world” graphs of up to ~125k nodes).”
>
> We agree with the reviewer that [1] is a relevant work to our work and included it in the citation list. Notice that one difference between [1] and DIMES is that [1] is only trained on TSP-100 and evaluated on larger graphs, while DIMES can be directly trained on large graphs with the help of meta updates. and that DIMES achieves an optimality gap of 7.61% on TSP-500, while [1] achieved an optimality gap of ~ 10% on TSP-200). More importantly, [1] is only designed for CO problems that can be transformed into a sequential decoding problem (such as routing), while DIMES can be applied to other general CO problems (such as MIS). We will add a paragraph to discuss those points about [1] vs DIMES in the 10-page version of the paper.
>
> > 4. “The attention model of Kool et al [34] is used as an RL baseline. However, this approach has been significantly improved in POMO (https://arxiv.org/abs/2010.16011) and then subsequently adapted for active search at test time (https://arxiv.org/abs/2106.05126).”
> We run the POMO+EAS code by Hottung et al. with the POMO model trained on TSP100. The experiments ran on a 16GB GPU with batch size 1. Here are the results:
>
> | Setting | TSP500     | TSP1000| TSP10000|
> | ----------- | ----------- | ----------- | ----------- |
> | POMO+EAS-Emb     | 19.24       | OOM | OOM|
> | POMO+EAS-Lay | 19.35       | OOM | OOM|
> | POMO+EAS-Tab | 24.54       | 49.56 | OOM|
> | DIMES | 16.84       | 23.69 | 74.06|
>
> where “OOM” indicates “out of GPU memory”. The results suggest that POMO+EAS may not work well on larger sizes due to distribution shift and high memory consumption. We have added those details in Table 1 of the paper.

---

> > ### Author Response · Authors · 2022-08-02
> > **Response (II)**
> >
> > > 5. “The authors only rerun one of the presented baselines (Att-GCN) on their hardware, so timing results for other baselines taken from Fu et al are potentially misleading.”
> >
> > Due to the time limit and unreproducibility of some baselines, we mainly report baseline timing results from Fu et al. However, since Att-GCN (Fu et al.,) significantly outperforms all existing baselines in the large-scale setting, we believe such a comparison can still highlight the value of our work.
> >
> > Besides, we have tried our best to ensure a fair comparison. Note that only MCTS is evaluated on CPU, while other parts of Att-GCN and DIMES, as well as other learning-based methods, are evaluated on GPU. In our experiments, we used the same GPU (1080 Ti) as Fu et al., so comparing the time of our GPU experiments with those in Fu et al. is already fair. However, we were not able to find the same CPU as that of Fu et al. So, we re-ran the MCTS of Att-GCN on our CPU to ensure a fair comparison between the MCTS times of DIMES and Att-GCN.
> >
> > > 6. “In Table 3, why is LwD with sampling (RL+S) not included? As LwD is better than DIMES in greedy mode, but DIMES is better with sampling, would LwD with sampling not be expected to be best of all?”
> >
> > The LwD method also evaluated their models with sampling, as mentioned in “To this end, we evaluate algorithms on ER-[400, 500] and SATLIB datasets with varying numbers of samples or time limits.” of the LwD paper. We now have fixed its keyword in Table 1, listing LwD as an RL+Sampling method instead of RL+Greedy.
> >
> > > 7.  Regarding the fine-tuning of DIMES in table 1: (i) Is the fine-tuning time included in the run time (I would have expected so, but then I am surprised how, for example, DIMES is faster the AM with equivalent settings if it has to be fine-tuned first)? (ii) Which elements of the model are fine-tuned (as four variants are shown in Table 2(b) it is clear there is some choice to be made)?
> >
> > We thank the reviewer for the great question. The runtime of DIMES should be total fine-tuning time + total MCTS time. However, we mistakenly reported average fine-tuning time per instance + total MCTS time. We have fixed it in the revised version of the paper and report the results of both un-finetuned (w/o active search) and finetuned (w/ active) models.
> >
> > > 8.  “If all baselines were taken from Fu et al, can you comment on whether the training of these was a fair comparison to DIMES? For example, DIMES is trained on the target problem sizes but it is common to present results on larger TSP instances of models trained on smaller problems.”
> >
> > A main advantage of DIMES is that it can scale to large-scale graphs. On the other hand, we agree with you that comparing DIMES with other less-scalable methods on small graphs is also informative.  So, we report the additional results of DIMES on TSP-100 in Appendix F.1.  We can see that DIMES outperforms the state-of-the-art Att-GCN in terms of the optimality gap (0.0103% v.s. 0.0370%). In fact, the performance of both models are already very close to the optimal solutions.
> >
> > > 9. “In the meta-learning ablations, why would training the network using meta-learning (targeted to give good performance after T steps of fine-tuning on a specific instance), give better performance when T=10 during training and T=0 at inference (i.e why is row 2 of Table 2a better than row 1?)”
> >
> > Thanks for the insightful question. We have a hypothesis but not sure:
> >
> > In case I (training T=0 and inference T=0), since the optimal probability distribution should be 1 for the optimal solution and 0 for other solutions, the softmax operation will require the near-optimal parameterization $\theta$ to have large output values, which is difficult to be learned or produced by a Lipschitz-continuous neural network.
> >
> > In case II (training T=10 and inference T=0), the model is using an inner optimization (i.e., T inner steps during training) in the objective and does not need to produce the large values of $\theta$, because the inner gradient updates will push small values produced by the neural network to large values. Hence, meta updates (with training T=10) help regardless of whether fine-tuning is used in the evaluation.
> >
> > > 10. Why is S2V-DQN not used as a baseline in Table 3, when it is stated to be a baseline on line 297?
> >
> > This is a typo. We have removed it from the paper.
> >
> > [1] Drori, Iddo, et al. "Learning to solve combinatorial optimization problems on real-world graphs in linear time." 2020 19th IEEE International Conference on Machine Learning and Applications (ICMLA). IEEE, 2020.

---

> ### Comment · Reviewer_fe3B · 2022-08-04
> **Thank you for the clarifications**
>
> I thank the reviewers for their detailed responses.  I’ve commented on these point by point below, but in summary, I still retain some reservations regarding DIMES novelty claims regarding “heatmaps” vs “continuous distributions”, and the way in which baselines are chosen/discussed.  With that said, the most significant concern I had (that the main results table simply presented unfeasible timings) has been addressed.  Therefore I am happy to increase my score to a 5, as the work does present interesting ideas and I now trust the results are valid/presented in good faith.
>
> 1.  I’m afraid this is still not clear to me.  My understanding is that, taking the definition of the sampling distribution for the next node in a TSP tour (eq. 8), if you are at node $i$, the probability of going to node j next is $\propto \mathrm{exp}(\theta_{i,j})$.  The “autoregression” (dependence on the nodes selected at positions 1,…,i-1) only appears as (i) masking the already visited nodes (invalid actions) and (ii) normalising the overall sampling distribution.  Both of these dependencies are trivial and don’t amount to meaningful insights (indeed, surely they must also be used when sampling from heatmaps in prior works, otherwise the tours generated would not be valid/distribution would be un-normalised).  Whilst you contrast your approach to heatmaps as "the values in the heatmap correspond to the estimated probability of each edge to be included in the optimal solution” in the rebuttal, the paper says of DIMES “ the higher valued $\theta_{i,j}$ means the higher probability for the edge from node $i$ to node $j$ to be sampled”.  I’m more than happy for my misunderstanding to be clarified, but right now I stand by my original view.
>
> 2. Understood.  I would note that perhaps deleting the second comma in: “To our knowledge, we are the first to apply meta-learning over a collection of CO problem instances, where each instance graph is treated as one of a collection tasks a sub-task in a unified framework.” would make this more clear (as Zhang et al are also “applying meta-learning to a collection of problem instances”).
>
> 3. I’m glad the extra context will be included in the revision.  However, I don’t see why Driori et al is more limited in scope than DIMES as the authors claim.  Both are decoded autoregressivley (whereas DIMES uses the masked “compact parameterisation” to encode the action probs given a trajectory + node, Driori et al uses masked dot-product attention).  Moreover, without further details on the training costs of baselines, comparing DIMES fine-tuned on larger instances to the generalisation performance of other methods does not strictly show DIMES to be a better approach.
>
> 4. Thank you for these results, which are impressive and present DIMES in a positive light.  However, again, they conflate generalisation ability with performance.  I don’t question that it is great DIMES can be trained on big instances and meta-learn for even more performant policies.  However, the paper is comparing DIMES to the performance of models trained on smaller TSP problems without sufficient context for how large of a disadvantage this is.  To be concrete, Table 1 has DIMES with some greedy decoding (18.93 on TSP-500) beating larger Transformer-based models (e.g. AM) and even the TSP active search SOTA (POMO-EAS).  I would strongly expect that if these models were trained on TSP-500 instances, they would be highly competitive with DIMES.  None of this is to criticize DIMES — of course, the feasibility of training is important — but I do think that the current sentence added to address this “Note that baselines are trained on small graphs and evaluated on large graphs, while DIMES can be trained directly on large graphs.” is insufficient as it really could be argued that none of the learning baselines in Table 1 are a fair direct comparison (e.g. could the models be trained for the same amount of time and then evaluated?).
>
> 5. I understand the challenge — my primary reason for suspicion was the odd timings discussed in your response to 7, with your clarification on that point I am happy to accept the timings are presented in good faith.
>
> 6. Understood — thank you.
>
> 7. Ah wow — that makes a big difference but looks much more reasonable now!
>
> 8. I appreciate the additional results, however my point was (as elaborated in response 4) that table 1 could be mis-leading as it compares generalisation to direct performance.
>
> 9. I am grateful to the authors for addressing my curiosity and certainly this is not a critical point with respect evaluating the paper.
>
> 10.  Understood — thank you.

---

> > ### Author Response · Authors · 2022-08-08
> > **Further Response**
> >
> > We thank the reviewer for their constructive feedback. Here’re our responses to the reviewer’s further comments.
> >
> > As for point 1: The main difference between previous “heatmaps” and our “continuous parameterization” is that each value in a previous “heatmap” is the **marginal probability** of an edge to be included in the optimal solution, while in our model  $\theta_{i,j}$ is the **conditional probability** of the next node ($i$) conditioned on the (embedded) partial solution so far (i.e., we use a Markov-chain of order k >= 1). We hope this would make the distinction clearer.
> >
> > As for point 2: Thanks for your comment. We will adopt your suggestion.
> >
> > As for points 3, 4, and 8: To make our experimental results more informative, we further evaluate the performance of DIMES trained on TSP-100 and evaluated on larger graphs.
> >
> > | Method | Type | TSP-500 | TSP-1000 | TSP-10000 |
> > |-|-|-|-|-|
> > | LKH-3 | OR | 16.55\* | 23.12\* | 71.77\* |
> > | EAN | RL+S | 28.63 | 50.30 | OOM |
> > | EAN | RL+S+2-OPT | 23.75 | 47.73 | OOM |
> > | AM | RL+S | 22.64 | 42.80 | 431.58 |
> > | AM | RL+G | 20.02 | 31.15 | 141.68 |
> > | AM | RL+BS | 19.53 | 29.90 | 129.40 |
> > | GCN | SL+G | 29.72 | 48.62 | OOM |
> > | GCN | SL+BS | 30.37 | 51.26 | OOM |
> > | POMO+EAS-Emb | RL+AS | 19.24 | OOM | OOM |
> > | POMO+EAS-Lay | RL+AS | 19.35 | OOM | OOM |
> > | POMO+EAS-Tab | RL+AS | 24.54 | 49.56 | OOM |
> > | Att-GCN | SL+MCTS | 16.97 | 23.86 | 74.93 |
> > | DIMES trained on TSP-n | RL+S | 18.84 | 26.36 | 85.75 |
> > | DIMES trained on TSP-100 | RL+S | 19.21 | 27.21 | 86.24 |
> >
> > We can see that the performance of DIMES does not drop too much. One of our hypotheses is that the graph sparsification schema in our neural network (See appendix C.1) avoids the explosion of activation values in the graph neural network. Another hypothesis is that meta learning tends to not generate too extreme values in  $\theta$ (see point 9 of our previous response) and hence improve the generalization capability.
> >
> > One may wonder how to make the previous methods scale up to TSP-500/1000/10000, and if such scaling were successful, what would be the relative performance of those methods compared to DIMES? Answering these questions requires further research, and is beyond the scope of this paper.

---

> > > ### Public Comment · ~Yi_Ju1 · 2024-02-21
> > > **on understanding $\theta$**
> > >
> > > I am still very confused on your interpretation of $\theta_{ij}$: "the conditional probability of the next node (i) conditioned on the (embedded) partial solution so far". If this is the case, what does $j$ mean? Does $j$ refer to a specific partial solution so far?

---

> > > > ### Public Comment · ~Ruizhong_Qiu1 · 2024-02-21
> > > > **The meaning of $\theta_{i,j}$**
> > > >
> > > > Thanks for your interest in our work. For TSP, $i$ and $j$ are two nodes. To be more specific, $\theta_{i,j}$ means the logit for the conditional probability that the next node is $j$ if we are currently at node $i$, as defined in Eq. (10) in our paper. For example, if there are $n=5$ nodes, and the partial solution is $(3,1,5)$, then the probability that the next node is $4$ is $\frac{\exp(\theta_{5,4})}{\exp(\theta_{5,2})+\exp(\theta_{5,4})}$. Please feel free to let us know if you have any further questions.

---

### Official Review · Reviewer_dJ85 · 2022-07-09

**Rating:** 6
**Confidence:** 4
**Soundness:** 2 fair
**Presentation:** 3 good
**Contribution:** 3 good

**Summary:**

This work proposes a novel DIMES framework (stands for differentiable meta solver) to tackle large-scale learning-based combinatorial optimization problems. The key novelties and contributions are 1) an RL-based approach to train the widely-used GCN model to generate probability heatmap, and 2) a meta-learning approach to further finetune the solutions at inference. Experimental results show that the proposed DIMES can achieve promising performance for large-scale TSP and MIS problems with up to 10,000 nodes.


**Questions:**

- DIMES needs to use RL to directly train deep GNN for large-scale problem instances with up to 100,000 nodes. Is there any challenge for such training? How long will DIMES take to converge for TSP500/1000/10000 (#training instances and wall-clock time)?

- TSP/MIS is a good testbed for neural combinatorial optimization. But the real-world applications will typically have problems with various structures that can not be solved by classical solvers. This is an important motivation for learning-based solvers without domain knowledge. Can DIMES generalize to other routing problems such as those in the AM paper?

- Citation [42] and [43] in the paper are for the same AM paper but with two different years (the 2019 one is correct).


**Limitations:**

Yes, the limitations have been adequately addressed in Section 5 Concluding Remarks. I do not see any potential negative societal impact of this work.

**Strengths And Weaknesses:**

**Strengths:**

+ Learning to solve large-scale combinatorial optimization problems is crucial for many real-world applications. This work is a timely contribution to an important research topic.

+ To my understanding, the proposed RL-based approach for training GCN to obtain the probability heatmap is novel. It is quite promising to see a pure end-to-end RL-based approach can tackle TSP with up to 10,000 nodes.

+ The meta-learning based fine-tuning strategy is also new for neural combinatorial optimization.

+ The experimental results on large-scale TSP are good.

**Weaknesses:**

I cannot give a clear acceptance to the current manuscript due to the following concerns:

**1. Inaccurate Contribution:** One claimed contribution of this work is the compact continuous parameterization of the solution space. However, as discussed in the paper, DIMES directly uses the widely-used GNN models to generate the solution heatmap for TSP[1,2] and MIS[3] problems, respectively. The credit for compact continuous parameterization should be given to the previous work [1,2,3] but not this work.

For TSP, Joshi et al.[1] have systemactilly studied the effect of different solution decoding (e.g., Autoregressive Decoding (AR) v.s. Non-autoregressive decoding (NAR, the heatmap approach) and learning methods ( supversied learning (SL) v.s. reinforcement learning (RL)). To my understanding, the combination of AR + SL, AR + RL and NAR(heatmap) + SL have been investigated in Joshi et.al. and other work (e.g., PtrNet-SL, PtrNet-RL/AM, GCN), but I am not aware of othe work on NAR(heatmap) + RL. The NAR + RL combination could be the novel contribution of this work.

**2. Actual Cost of Meta-Learning:** The meta-learning (meta-update/fine-tuning) approach is crucial for the proposed method's promising performance. However, its actual cost has not been clearly discussed in the main paper. For example, Table 1 reports that DIMES only needs a few minutes to solve 128 TSP500/TSP1000 and 16 TSP10000 instances. However, at inference, DIMES actually needs extra meta-gradient update steps to adapt its model parameters to each problem instance. The costs of the meta-gradient steps are 1.5h - 10h for TSP500 to TSP10000 as reported in Appendix C.1. Since all the other heuristic/learning methods do not require such meta update step, it is unfair to report that the runtime of DIMES is only a few minutes (which should be a few hours) in Table 1.

**3. Generalization v.s. Testing Performance:** To my understanding, all the other learning-based methods in Table 1 are trained on TSP100 instances but not TSP500-TSP10000 as for DIMES. Therefore, the results reported in Table 1 are actually their out-of-distribution generalization performance. There are two important generalization gaps compared with DIMES: 1) generalization from TSP100 to TSP10000, 2) generalization to the specific TSP instances (the fine-tuning step in DIMES). I do see it is DIMES's own advantages (direct RL training for large-scale problems + meta fine-tuning) to overcome these two generalization gaps, but the difference should be clearly clarified in the paper.

In addition, it is also interesting to see a comparison of DIMES with other methods on TSP100 (in-distribution testing performance) with/without meta-learning.

**4. Advantage of NAR(heatmap) + RL + Meta-Learning:** From Table 1&2, for TSP1000, the generalization performance of AM (G: 31.15, BS: 29.90) trained on TSP100 is not very far from the testing performance of DIMES without meta-learning (27.11) directly trained on TSP1000. It could be helpful to check whether the more powerful POMO approach[4] can have a smaller performance gap. Reporting the results for POMO and DIMES without meta-learning for all instances in Table 1 could make the advantage of the NAR(heatmap) + RL approach in DIMES much clearer.

Hottung et al.[5] shows that POMO + Efficient Active Search (EAS) can achieve promising generalization performance for larger TSP instances on TSP and CVRP. The comparison with POMO + EAS could be important to better evaluate the advantage of meta-learning in DIMES.

[1] Chaitanya K Joshi, Quentin Cappart, Louis-Martin Rousseau, Thomas Laurent, and Xavier Bresson. Learning tsp requires rethinking generalization. arXiv preprint arXiv:2006.07054,2020.

[2] Chaitanya K Joshi, Thomas Laurent, and Xavier Bresson. An efficient graph convolutional network technique for the travelling salesman problem. arXiv preprint arXiv:1906.01227, 2019.

[3] Zhuwen Li, Qifeng Chen, and Vladlen Koltun. Combinatorial optimization with graph convolutional networks and guided tree search. NeurIPS 2018.

[4] Yeong-Dae Kwon, Jinho Choo, Byoungjip Kim, Iljoo Yoon, Youngjune Gwon, and Seungjai Min. POMO: Policy optimization with multiple optima for reinforcement learning. NeurIPS 2020.

[5] André Hottung, Yeong-Dae Kwon, and Kevin Tierney. Efficient active search for combinatorial optimization problems. ICLR 2022.

---

> ### Author Response · Authors · 2022-08-02
> **Response (I)**
>
> We greatly appreciate the enthusiasm and interest the reviewer has shown towards our work. We have provided our responses below.
>
> > 1. "Inaccurate Contribution: One claimed contribution of this work is the compact continuous parameterization of the solution space. However, as discussed in the paper, DIMES directly uses the widely-used GNN models to generate the solution heatmap for TSP[1,2] and MIS[3] problems, respectively. The credit for compact continuous parameterization should be given to the previous work [1,2,3] but not this work.”
>
> The heatmaps in previous works are generated in a fully non-autoregressive manner, where the values in the heatmap correspond to the estimated probability of each edge to be included in the optimal solution. In our work, the n-by-n matrix $\theta$ is used to parameterize distribution $q$, in an autoregressive manner,  in estimating the probability of the next node conditioned on the path (partial solution) so far. That is, the elements in our $\theta$ matrix are not the estimated probabilities of edges. In other words, although the heatmaps in previous work look like our $\theta$ syntactically (as both are in the form of an n-by-n matrix), the semantics behind them are fundamentally different. To avoid possible confusion, we have revised our paper to make the distinction between “heatmap” in previous work and our novel approach clearer, by calling our $\theta$ matrix “continuous parameterization” instead of “heatmap”.
>
> > 2. "Actual Cost of Meta-Learning: The meta-learning (meta-update/fine-tuning) approach is crucial for the proposed method's promising performance. However, its actual cost has not been clearly discussed in the main paper.”
> The time 1.5h - 10h of DIMES is training time (i.e., time of meta-updates) on training instances (not on test examples), which should not be in Table 1. The evaluation runtime of DIMES in Table 1 consists of fine-tuning steps and MCTS on test instances. Therefore, it is a fair comparison to previous work.
>
> However, we find that we report the wrong runtime for our DIMES model by mistake. The runtime of DIMES should be total fine-tuning time + total MCTS time. But we mistakenly reported average fine-tuning time per instance + total MCTS time. We have fixed it in the revised version of the paper and report the results of both un-finetuned (w/o active search) and finetuned (w/ active) models.
>
> > 3. "Generalization v.s. Testing Performance: To my understanding, all the other learning-based methods in Table 1 are trained on TSP100 instances but not TSP500-TSP10000 as for DIMES. Therefore, the results reported in Table 1 are actually their out-of-distribution generalization performance.” & “In addition, it is also interesting to see a comparison of DIMES with other methods on TSP100 (in-distribution testing performance) with/without meta-learning.”
>
> You are right: for other DRL methods in Table 1, they are trained on TSP-100 and evaluated on TSP-500, 1000, and 1000. We agree a direct comparison of DIMES and other DRL methods on TSP-100 is informative. The detailed comparison is included in Appendix F.1 of the paper. We can see that DIMES outperforms the state-of-the-art Att-GCN in terms of the optimality gap (0.0103% v.s. 0.0370%). However, the performance of both models are already very close to the optimal solutions.
>
> Besides, we added into Section 4.1.2 Line 241-243 a clarification that previous methods are trained on small graphs while DIMES is trained on large graphs.
>
> > 4. "Hottung et al.[5] shows that POMO + Efficient Active Search (EAS) can achieve promising generalization performance for larger TSP instances on TSP and CVRP. The comparison with POMO + EAS could be important to better evaluate the advantage of meta-learning in DIMES.”
>
> We run the POMO+EAS code by Hottung et al. with the POMO model trained on TSP100. The experiments ran on a 16GB GPU with batch size 1. Here are the results:
> | Setting | TSP500     | TSP1000| TSP10000|
> | ----------- | ----------- | ----------- | ----------- |
> | POMO+EAS-Emb     | 19.24       | OOM | OOM|
> | POMO+EAS-Lay | 19.35       | OOM | OOM|
> | POMO+EAS-Tab | 24.54       | 49.56 | OOM|
> | DIMES | 16.84       | 23.69 | 74.06|
>
> where “OOM” indicates “out of GPU memory”. The results suggest that POMO+EAS may not work well on larger sizes due to distribution shift and high memory consumption. We have added those details in Table 1 of the paper.

---

> > ### Author Response · Authors · 2022-08-02
> > **Response (II)**
> >
> > > 5. “DIMES needs to use RL to directly train deep GNN for large-scale problem instances with up to 100,000 nodes. Is there any challenge for such training? How long will DIMES take to converge for TSP500/1000/10000 (#training instances and wall-clock time)?”
> >
> > In this paper, we only tried on graphs with up to 10,000 nodes, while we do not see obvious challenges to apply DIMES to larger graphs (e.g., 100,000 nodes).
> > The time 1.5h - 10h of DIMES is training time (i.e., time of meta-updates) on training instances (not on test instances). Please see Appendix D.1 for the details.
> >
> > > 6. “TSP/MIS is a good testbed for neural combinatorial optimization. But the real-world applications will typically have problems with various structures that can not be solved by classical solvers. This is an important motivation for learning-based solvers without domain knowledge. Can DIMES generalize to other routing problems such as those in the AM paper?”
> >
> > The generality of our unified framework is based on the assumption that each feasible solution of the CO problem on hand can be represented with a vector of 0/1 valued variables (typically corresponding the selection of a subset of nodes or edges), which is fairly mild and generally applicable to many CO problems beyond MIS and (see Karp's 21 problems) with few modifications. The design principle of auxiliary distribution is to design an autoregressive model that can sequentially grow a partial solution toward a valid complete solution.  This design principle is also proven to be generic enough for many problems in neural learning, including CO solvers.
> > As for problems beyond this assumption (as discussed in Section 5), Mixed Integer Programming (MIP) is an example, where the variables can take multiple integer values instead of binary.  However, such a limitation can be addressed by reducing each integer value (x) to a sequence of bits (log x) and by predicting the bits one after another. In other words, a multi-valued MIP problem can be reduced to a binary-valued MIP problem with more variables, as is shown in [1].
> > [1]: Solving Mixed Integer Programs Using Neural Networks
> >
> > > 7. "Citation [42] and [43] in the paper are for the same AM paper but with two different years (the 2019 one is correct)."
> >
> > Thanks for pointing it out. We have fixed this.

---

> > > ### Comment · Reviewer_dJ85 · 2022-08-07
> > > **Follow-up Questions**
> > >
> > > Thank you for your thorough response. I've also read other reviewers' comments and have some follow-up questions.
> > >
> > > **1. Heatmap v.s. Continuous Parameterization**
> > >
> > > Thank you for the clarification. However, the proposed method uses exactly the same models proposed in previous works for TSP/MIS, which have already developed the compact n*n output. I can understand your claim that the interpretation of the output matrix could be different due to the different decoding methods. But since the compact output matrix structure is from previous work, I still think the main contribution of "introduces a compact continuous space for parameterizing the underlying distribution of candidate solutions" is overclaimed.
> > >
> > > **2/5. Run Time and Training Time**
> > >
> > > Now the runtime reported in Table 1 makes sense.
> > >
> > > A follow-up question is about the extremely fast training time for DIMES (1.5h - 10h). In [1], the authors report that training on TSP200 could be extremely challenging, which did not converge after 500 hours. The POMO paper [2] reports that it needs one week (168 hours) to observe full converge on TSP100. Why does DIMES only need 1.5h - 10h as the training time for much larger problems such as TSP500/1000/10000?
> > >
> > >
> > > Reference
> > >
> > > [1] Chaitanya K Joshi, Quentin Cappart, Louis-Martin Rousseau, Thomas Laurent, and Xavier Bresson. Learning tsp requires rethinking generalization. arXiv preprint arXiv:2006.07054,2020.
> > >
> > > [2] Yeong-Dae Kwon, Jinho Choo, Byoungjip Kim, Iljoo Yoon, Youngjune Gwon, and Seungjai Min. POMO: Policy optimization with multiple optima for reinforcement learning. NeurIPS 2020.

---

> > > > ### Author Response · Authors · 2022-08-08
> > > > **Further Response**
> > > >
> > > > We thank the reviewer for their insightful follow-up questions, and here are our responses.
> > > >
> > > > **As for point 1:** The main difference between previous “heatmaps” and our “continuous parameterization” is that each value in a previous “heatmap” is the *marginal probability* of an edge to be included in the optimal solution, while in our model  $\theta_{i,j}$ is the *conditional probability* of the next node ($i$) conditioned on the (embedded) partial solution so far (i.e., we use a Markov-chain of order k >= 1). We hope this would make the distinction clearer. Therefore, while the compact output matrix structure is the same, the probability models (which we argue is not simply interpretation) are different.
> > > >
> > > > **As for point 2:** According to our experimental observation, there are three factors that seem to contribute to the fast training of DIMES:
> > > >
> > > > a) Meta-learning helps DIMES to have a more stable training process.
> > > >
> > > > As is shown in Figure 3 in [1], the stability of training seems fairly related to the convergence performance. In our experiments, the loss curve of DIMES is more stable, which may explain its fast convergence. Moreover, the stability of training enables us to use a larger learning rate. (Our learning rate is 1e-3, while that of AM and POMO is 1e-4.) This also accelerates training.
> > > >
> > > > Although we do not have a theoretical explanation, we have a conjecture: the inner optimization in meta-learning helps the NN not to produce extreme values of $\theta$ and hence improves stability. Since the optimal probability distribution should be 1 for the optimal solution and 0 for other solutions, the softmax operation will require the near-optimal parameterization $\theta$ to have large values. When there is no inner optimization, the model has to produce large values of $\theta$, so training the model is more likely to be unstable. When using inner optimization, the model does not need to produce large values of $\theta$, because inner optimization can help push small values of $\theta$ to large values.
> > > >
> > > > b) The loss function of DIMES seems to have lower variance, making DIMES more sample-efficient.
> > > >
> > > > Here is a comparison among AM, POMO, and DIMES:
> > > >
> > > > | Method | AM | POMO | DIMES |
> > > > |-|-|-|-|
> > > > | Training scale | TSP-100 | TSP-100 | TSP-500/1000/10000 |
> > > > | Total descent steps of GNN | 250,000 | 312,600 | 120/120/50 |
> > > > | Batch size | 512 | 64 | 3 |
> > > > | Total training instances | 128,000,000 | 20,000,000 | 360/360/150 |
> > > > | Training GPUs | 2 | 1 | 1 |
> > > > | Per-step training time | 0.66s | ~0.28s | ~45s/51s/12m |
> > > > | Total training time | ~46h | ~1 day | ~1.5h/1.7h/10h |
> > > >
> > > > (Note: Per-step training time is calculated based on total training time reported in the papers.)
> > > >
> > > > The table shows that DIMES is more sample-efficient than AM/POMO, achieving stable training using only 3 instances per meta-gradient descent step. Hence, its total training time is accordingly much shorter, even though its per-step time is longer.
> > > >
> > > > c) The sampling scheme of DIMES is more scalable, which requires less GNN computation than AM/POMO.
> > > >
> > > > AM/POMO requires $n$ times of GNN computation to sample a solution on TSP-$n$, because it has to update context embedding at each autoregressive step. For DIMES, it needs GNN computation only once to compute $\theta$ in a non-autoregressive manner, and then it autoregressively samples solutions using $\theta^{(T)}$ without GNN re-computation. Hence, the sampling scheme of DIMES is more scalable. Besides, we implement the sampling procedures in C++ LibTorch to speed up further.
> > > >
> > > > We will add a figure on training dynamics to the revised version of the paper to make the information more complete.
> > > >
> > > > [1] Kwon et al. POMO: Policy Optimization with Multiple Optima for Reinforcement Learning

---

> > > > > ### Comment · Reviewer_dJ85 · 2022-08-10
> > > > > **Thank You**
> > > > >
> > > > > Thank you for your further response.
> > > > >
> > > > > For point 1, I can understand the authors' viewpoint on the contribution, but still think a new decoding method over the same compact output matrix (and indeed the whole model structure) cannot fully support the claim that it "introduces a compact continuous space for parameterizing the underlying distribution of candidate solutions".
> > > > >
> > > > > For point 2, thank you for the explanation, and the extremely fast training time is quite surprising and promising. I agree with the authors that adding this discussion and the figure on training dynamic to the revised paper could be very helpful.
> > > > >
> > > > > I raise my score to 6.

---

### Official Review · Reviewer_LJJE · 2022-07-10

**Rating:** 6
**Confidence:** 5
**Soundness:** 3 good
**Presentation:** 2 fair
**Contribution:** 3 good

**Summary:**

This paper studies a differential meta solver for CO problems and its performance is demonstrated by using TSP instances (size 500, 1000, and 10000). The difference between the proposed method in this paper and existing DRL-based methods is from the fact that DIMES focuses on a parameterized continuous space to represent solutions of CO problems. Experimental results indicate that DIMES show better performance than DRL-based solvers. Ablation studies on DIMES show that the proposed components work effectively.

**Questions:**

- I feel that the proposed concept is interesting to investigate, and I'm interested in how to design auxiliary functions (this is also noted in sec.5 by the authors). I also read proofs in the supplemental material. I wonder if we can design such functions for various CO problems. The paper proposed the answer for TSP and MIS, but defining such functions (with proofs) seems to be challenging (as many variations on objective functions and constraints). If this is not straightforward (or this requires some hand-crafted works like parameter tuning), I feel that DIMES requires us to design such functions (this is similar to us designing good encoding-decoding scheme for DRL-based methods, good heuristics in OR, better algorithms, etc.).

- In Table 1, LKH-3 with TSP-1000 requires 38.09[m] (this may be from Fu et al. [16] https://arxiv.org/pdf/2012.10658.pdf). Thus, it seems that the DIMES outperforms the famous heuristic solver. However, for example in Table 1 [https://arxiv.org/pdf/1402.4699.pdf] (Note that I do not want to say like `this paper is good/bad`, I just cite experimental results from this.), LKH-3 seems to solve TSP instances (size from 9847 to 16862) efficiently (from 381[s] to 976[s]). I wonder what is the differences between such traditional benchmark TSP instances (like ja9847) and those generated by Fu et al. [16] (TSP-1000/TSP-10000). This behavior is also seen in existing papers like AM's paper (LKH3 requires 21[m] to solve TSP-100). Such results (often reported in papers proposing learning-based solvers) seem to be strange to me. Could you please explain them?

- For datasets, the authors mentioned two sources from [35] (train) and [16] (test). The difference and similarities of datasets among them should be explained.

- Related to the previous question, I'm interested in the learned results that can be generalized to other instances. For example, TSP-1000 with MAML requires 1.7[h] learning times + 4.47[m] to generate solution (1.7[h] from supp mat and 4.47[m] from Table 1.), but LKH-3 requires 38.09[m]. So, if the learning results are not shareable with other kinds of datasets, the learning process is a bit resource-consuming trial. Do you have any ideas or findings?

- The dimension T (of MAML) is fixed. Is this tunable? (or need to be tuned to get better solutions?). In the ablation study (c), it seems that learning more by MAML produces better solutions. Is this interpretation true?

- (L.45) Instead of previous DIR-based CO sovles; what is DIR-based? sovles => solves. Some other english errors should be updated for readability.


**Limitations:**

After reading the main paper and appendix, I feel that the authors carefully discuss the negative impacts.

**Strengths And Weaknesses:**

[Strength]
- Focusing on the continuous space (Eq.2) could widen this research field (rather than the encoding-decoding scheme in DRL-based methods).
- Large instances (e.g., TSP-10000) are tackled by DIMES (Note that such scalability could be hard for DRL-based methods in the current status).

[Weakness]
- The generalization ability when designing auxiliary distributions is unclear.
- The discussion and background of MAML (e.g., the idea of T gradient updates) are a bit hard to follow.

---

> ### Author Response · Authors · 2022-08-02
> **Response (I)**
>
> We appreciate the in-depth questions and suggestions given by the reviewer. We have provided our responses below.
> > 1. “The generalization ability when designing auxiliary distributions is unclear.” & “I feel that the proposed concept is interesting to investigate, and I'm interested in how to design auxiliary functions (this is also noted in sec.5 by the authors). I also read proofs in the supplemental material. I wonder if we can design such functions for various CO problems…”
>
> The generality of our unified framework is based on the assumption that each feasible solution of the CO problem on hand can be represented with a vector of 0/1 valued variables (typically corresponding the selection of a subset of nodes or edges), which is fairly mild and generally applicable to many CO problems beyond MIS and TSP (see Karp's 21 problems) with few modifications. The design principle of auxiliary distribution is to design an autoregressive model that can sequentially grow a partial solution toward a valid complete solution.  This design principle is also proven to be generic enough for many problems in neural learning, including CO solvers.
> As for problems beyond this assumption (as discussed in Section 5), Mixed Integer Programming (MIP) is an example, where the variables can take multiple integer values instead of binary.  However, such a limitation can be addressed by reducing each integer value (x) to a sequence of bits (log x) and by predicting the bits one after another. In other words, a multi-valued MIP problem can be reduced to a few binary-valued MIP problems, as shown in [1].
> [1]: Solving Mixed Integer Programs Using Neural Networks
>
> > 2. The discussion and background of MAML (e.g., the idea of T gradient updates) are a bit hard to follow.
>
> We have added a short description of MAML to Section 3.3 (line 170-174) to make the background of MAML clearer.
>
> > 3. “LKH-3 seems to solve TSP instances (size from 9847 to 16862) efficiently (from 381[s] to 976[s]). I wonder what is the differences between such traditional benchmark TSP instances (like ja9847) and those generated by Fu et al. [16] (TSP-1000/TSP-10000)...”
>
> You are right. Following the previous work, we used the default parameters of LKH-3 for all experiments, where the number of max trials is 10000.
>
> In our new experiments, we find that LKH-3 with less trials still have strong results indeed. Especially, when decreasing the number of max trials to 500 for TSP500, 500 for TSP1000, and 250 for TSP10000 to reduce the running times of LKH-3 which match that of DIMES+MCTS, we got the following results of LKH-3:
> | Setting | TSP500     | TSP1000| TSP10000|
> | ----------- | ----------- | ----------- | ----------- |
> | LKH-3 (10000 max trails)     | 16.55       | 23.12 | 71.77|
> | LKH-3 (less trails) | 16.55       | 23.12 | 71.79|
> | DIMES | 16.84       | 23.69 | 74.06|
>
> We can see that LKH-3 achieves the same performance with less trials except for the large-scale TSP10000 task. This shows that expert-designed solvers with careful parameter tuning still can outperform learning-based methods. We have included these additional results in our paper.

---

> > ### Author Response · Authors · 2022-08-02
> > **Response (II)**
> >
> > > 4. “For datasets, the authors mentioned two sources from [35] (train) and [16] (test). The difference and similarities of datasets among them should be explained.”
> >
> > The training data used by both DIMES and [35] are randomly sampled uniformly from a Euclidean space on the fly during training time The only difference in the data by DIMES and [35] is the use of different random seeds.
> > The test data [16] is generated in the same way by DIMES and [35]. For a fair comparison, we directly took the test data of [16] for evaluation.
> >
> > > 5. “For example, TSP-1000 with MAML requires 1.7[h] learning times + 4.47[m] to generate solution (1.7[h] from supp mat and 4.47[m] from Table 1.), but LKH-3 requires 38.09[m]. So, if the learning results are not shareable with other kinds of datasets, the learning process is a bit resource-consuming trial. Do you have any ideas or findings?”
> >
> > The insight of applying learning algorithms to combinatorial optimization problems is that after training over a set of instance problems (graphs), the learned neural network can achieve better average performance than the algorithms without such learning. In other words, the learned neural network has the functionality of sharing the learned knowledge (search strategies based on the estimated $q$ distribution) across graphs, including those not seen in the training set.  Therefore, as long as the training instances and test instances are from the same underlying distribution and share some of factorized parameters, the learning algorithms should work well.  Since the training cost is off-line, a well-trained model can be faster (less resource-consuming) than untuned heuristic algorithms in the testing phase (which is the focus in Table 1).
> >
> > > 6. “The dimension T (of MAML) is fixed. Is this tunable? (or need to be tuned to get better solutions?). In the ablation study (c), it seems that learning more by MAML produces better solutions. Is this interpretation true?”
> >
> > As shown in our ablation study (Table 2c), with the number of inner gradient updates $T$ increasing, the testing-phase performance improves accordingly. However, this also consumes more training time. Hence, there is a trade-off between performance and training time in practice.
> >
> > > 7.   “what is DIR-based? solves => solves. Some other english errors should be updated for readability.”
> >
> > Thanks for pointing out the typos. We have fixed them.

---

> > > ### Comment · Reviewer_LJJE · 2022-08-08
> > > **Thank you for your detailed comments.**
> > >
> > > I appreaciate the detailed comments from the authors.
> > >
> > > Although I concern the generalization ability when I review the paper, I can understand what the authors stated at (1), and then I think that the proposed framework has a kind of generalization ability (it is still not sure which typs of problems could fit).
> > >
> > > I also appreciate comments and updates (2)-(4), and (6)-(7). They increased the readability of the paper.
> > >
> > > In terms of (5), I think we have two types of generalization abilities for learning-based heuristics in the literature. In some NN-based studies, they discussed size-related property (learning NNs on TSP-50, and using learned NNs for different sizes like TSP-100). However, for optimization problems, we have a different type; how to generate instances  (for example, TSP-50 with different location distributions). I think the word _distributions_ have wider meanings, so I recommend the authors to clarify them if possible.
> > >
> > > Anway, thanks again for the comments with experimental results, which clarify my concerns.

---

### Official Review · Reviewer_fQdp · 2022-07-11

**Rating:** 5
**Confidence:** 4
**Soundness:** 3 good
**Presentation:** 3 good
**Contribution:** 3 good

**Summary:**

This paper improves existing DRL-based CO methods in two terms. Firstly, the authors leverage a continuous probabilistic space for the solutions, leading to a REINFORCEMENT-based training method which is more efficient than previous Q-learning or Actor-Critic methods. Besides, a MAML-based meta-learning framework is proposed.

**Questions:**

1.  How many trials are configured for LKH-3? Can you decrease the number of trials so that LKH-3 can be comparatively faster than DIMES, and report the time and tour lengths?
2. In L149 the authors write: "we also no longer need costly MCMC-based sampling for optimizing our model". If sampling is not needed, how do you estimate the expected cost value after the neural network predicts q?
3. What is the application scope of the proposed method? Beyond routing and independent set, can you list some other types of problems that the proposed method can tackle and some that may be hard to handle?
4. Is there an ablation study for with/without meta-learning?

**Limitations:**

The limitations are addressed.

**Strengths And Weaknesses:**

**Strengths**
1. This paper is well-motivated by several important issues in existing DRL-based CO methods.
2. The proposed REINFORCEMENT-based method and the MAML-based meta-learning method seem sound.
3. The proposed method can handle larger-sized problems than existing DRL-based CO methods.

**Weaknesses**
1. This paper aims for developing a new general framework, but the current evaluation is only for two problems. Evaluating more problems of different natures (such as covering/matching problems) will make this paper more concrete and convincing.
2. There are certain implementation details that seem unclear to me and I am expecting the authors to answer (see "Questions").

**Typos**
1. Line 45: What is "DIR"-based CO solver?
2. Repeated citations: 29/30, 34/35

---

> ### Author Response · Authors · 2022-08-02
> **Response**
>
> We thank the reviewer for their time, insightful comments, and questions. We have provided our responses below.
>
> > 1. “How many trials are configured for LKH-3? Can you decrease the number of trials so that LKH-3 can be comparatively faster than DIMES, and report the time and tour lengths?”
>
> Following the previous work, we used the default parameters of LKH-3 for all experiments, where the number of max trials is 10000.
> Especially, when decreasing the number of max trials to 500 for TSP500, 500 for TSP1000, and 250 for TSP10000 to reduce the running times of LKH-3 which match that of DIMES+MCTS, we got the following results of LKH-3:
>
> | Setting | TSP500     | TSP1000| TSP10000|
> | ----------- | ----------- | ----------- | ----------- |
> | LKH-3 (10000 max trails)     | 16.55       | 23.12 | 71.77|
> | LKH-3 (less trails) | 16.55       | 23.12 | 71.79|
> | DIMES | 16.84       | 23.69 | 74.06|
>
> We can see that LKH-3 achieves the same performance with less trials except for the large-scale TSP10000 task. This shows that expert-designed solvers with careful parameter tuning still can outperform learning-based methods. We have included these additional results in our paper.
>
> > 2. “In L149 the authors write: "we also no longer need costly MCMC-based sampling for optimizing our model". If sampling is not needed, how do you estimate the expected cost value after the neural network predicts q?”
>
> We meant that we use autoregressive factorization with sampling from the auxiliary distribution which is faster than sampling with MCMC from the distribution defined by the energy function. We have modified our sentence (page 4 line 149) to make this point non-ambiguous.
>
> > 3. “What is the application scope of the proposed method? Beyond routing and independent set, can you list some other types of problems that the proposed method can tackle and some that may be hard to handle?”
>
> The generality of our unified framework is based on the assumption that each feasible solution of the CO problem on hand can be represented with a vector of 0/1 valued variables (typically corresponding the selection of a subset of nodes or edges), which is fairly mild and generally applicable to many CO problems beyond MIS and TSP (see Karp's 21 problems) with few modifications. The design principle of auxiliary distribution is to design an autoregressive model that can sequentially grow a partial solution toward a valid complete solution.  This design principle is also proven to be generic enough for many problems in neural learning, including CO solvers.
>
> As for problems beyond this assumption (as discussed in Section 5), Mixed Integer Programming (MIP) is an example, where the variables can take multiple integer values instead of binary.  However, such an limitation can be addressed by reducing each integer value (x) to a sequence of bits (log x) and by predicting the bits one after another. In other words, a multi-valued MIP problem can be reduced to a few binary-valued MIP problems, as shown in [1].
> [1]: Solving Mixed Integer Programs Using Neural Networks
>
> > 4. “Is there an ablation study for with/without meta-learning?”
>
> Please see Table 2 (a) and Section 4.1.3 for our ablation study on meta-learning.

---

> > ### Comment · Reviewer_fQdp · 2022-08-08
> > **Thank you.**
> >
> > Thank the authors for the detailed feedback. I will retain my borderline accept score.
> >
> > I have an additional suggestion: the time/objective in your tables seems kind of messy. Is it possible to update the results by adjusting the running time of most methods? I think it will serve as a better benchmark for following research works on TSP.

---

### Author Response · Authors · 2022-08-02
**General Response**

We thank all the reviewers for their precious time and insightful comments. We appreciate that the reviewers recognize our work as well-motivated (Reviewer fQdp, Reviewer fe3B), technically sound (Reviewer fQdp), scalable (Reviewer fQdp, Reviewer LJJE, Reviewer dJ85), and is a timely contribution (Reviewer dJ85). To improve the paper quality, we respond to the reviewers’ comment by making the following major revisions to the paper:

1. Following the reviewers’ suggestions, we include the results of LKH-3 with less trials and POMO + EAS as additional baselines in Table 1. We can see that DIMES is still a state-of-the-art learning method.

2. We add a comparison of DIMES and other methods on TSP-100 in Appendix F.1 of the paper. We can see that DIMES still outperforms the state-of-the-art Att-GCN in terms of the optimality gap (0.0103% v.s. 0.0370%). In fact, the performance of both models is already very close to the optimal solutions.

3. “On the novelty of our proposed continuous compact parameterization compared to heatmap method in previous work.”

The heatmaps in previous works are generated in a fully non-autoregressive manner, where the values in the heatmap correspond to the estimated probability of each edge to be included in the optimal solution. In our work, the n-by-n matrix $\theta$ is used to parameterize distribution $q$, in an autoregressive manner,  in estimating the probability of the next node conditioned on the path (partial solution) so far. That is, the elements in our $\theta$ matrix are not the estimated probabilities of edges. In other words, although the heatmaps in previous work look like our $\theta$ syntactically (as both are in the form of an n-by-n matrix), the semantics behind them are fundamentally different. To avoid possible confusion, we have revised our paper to make the distinction between “heatmap” in previous work and our novel approach clearer, by calling our $\theta$ matrix only “continuous parameterization” instead of “heatmap”.

4. We have added a short description of MAML to Section 3.3 (line 170-174) to make the background of MAML clearer.

---

### Meta-Review · Area_Chair_ZBfg · 2022-08-26

**Recommendation:** Accept
**Confidence:** Certain

**Metareview:**

This paper proposes a differentiable meta-solver applicable to large-scale combinatorial optimization. After a thorough discussion phase, all the reviewers are on the positive side of this paper. The reviewers appreciated the novelty of this paper and the importance of scaling neural combinatorial optimization for large-scale instances. Overall, I recommend acceptance for this paper.

However, the reviewers also showed concerns about the presentation of this paper. The gap between generalization and testing performance is not clearly discussed and the connection to prior works using continuous latent space should be clearly stated. Since scalability is an important issue, it would be useful to clear up time/objective comparison and unify experimental settings as suggested by Reviewer fQdp and fe3B.




**Award:**

No

---

### Decision · Program_Chairs · 2022-09-14

Accept